# Unifying GANs and Score-Based Diffusion as Generative Particle Models

**Jean-Yves Franceschi**
Criteo AI Lab, Paris, France
jycja.franceschi@criteo.com

**Mike Gartrell**
Criteo AI Lab, Paris, France
mike.gartrell@acm.org

**Ludovic Dos Santos**[*]
Criteo AI Lab, Paris, France
l.dossantos@criteo.com

**Thibaut Issenhuth**[*]
Criteo AI Lab, Paris, France
LIGM, Ecole des Ponts, Univ Gustave Eiffel,
CNRS, Marne-la-Vallée, France
t.issenhuth@criteo.com

**Emmanuel de Bézenac**[*]
Seminar for Applied Mathematics,
D-MATH, ETH Zürich, Rämistrasse 101,
Zürich-8092, Switzerland
emmanuel.debezenac@sam.math.ethz.ch

**Mickaël Chen**[*]
Valeo.ai, Paris, France
mickael.chen@valeo.com

**Alain Rakotomamonjy**[*]
Criteo AI Lab, Paris, France
a.rakotomamonjy@criteo.com

## Abstract

Particle-based deep generative models, such as gradient flows and score-based diffusion models, have recently gained traction thanks to their striking performance. Their principle of displacing particle distributions using differential equations is conventionally seen as opposed to the previously widespread generative adversarial networks (GANs), which involve training a pushforward generator network. In this paper we challenge this interpretation, and propose a novel framework that unifies particle and adversarial generative models by framing generator training as a generalization of particle models. This suggests that a generator is an optional addition to any such generative model. Consequently, integrating a generator into a score-based diffusion model and training a GAN without a generator naturally emerge from our framework. We empirically test the viability of these original models as proofs of concepts of potential applications of our framework.

## 1 Introduction

Score-based diffusion models (Song et al., 2021) have recently received significant attention within the machine learning community, due to their striking performance on generative tasks (Rombach et al., 2022; Ho et al., 2022). Similarly to gradient flows, these models involve systems of particles, where the displacement of the particle distribution is described by a differential equation parameterized by a gradient vector field. Such particle-based deep generative models are typically seen as opposed to generative adversarial networks (GANs, Goodfellow et al., 2014), as the latter involves adversarial training of a generator network (Dhariwal & Nichol, 2021; Song, 2021; Xiao et al., 2022).

In this paper, we challenge the conventional view that particle and adversarial generative models are opposed to each other. We make the following contributions.

**A unified framework.** We present a novel framework that unifies both classes of models, showing that they are based on similar particle evolution equations. Particle models follow a gradient vector

---

[*]Authors listed in a randomly chosen order.

37th Conference on Neural Information Processing Systems (NeurIPS 2023).

Table 1: Taxonomy of particle models, including our proposed hybrid models: Score GANs and Discriminator Flows.

| Model | Generator | Flow type $\nabla h_{\rho_t}$ |
|---|---|---|
| Wasserstein gradient flows | ✗ | Wasserstein gradient $-\nabla_W \mathcal{F}(\rho_t)$ |
| Stein gradient flows | ✓ | |
| Score-based diffusion models | ✗ | $\alpha_t \nabla \log\left[p_{\text{data}} \star k_{\text{RBF}}^{\sigma(t)}\right] - \beta_t \nabla \log \rho_t$ |
| Score GANs | ✓ | |
| Discriminator Flows | ✗ | $-\nabla\big(c \circ f_{\rho_t}\big)$ |
| GANs | ✓ | where $f_{\rho_t}$ is a discriminator between $\rho_t$ and $p_{\text{data}}$ |

field during inference; in a similar fashion, the generator's outputs can be seen as following the same gradient field during training, up to a specific smoothing due to the generator. Building upon prior scattered literature, we propose a new framework that encompasses a variety of methods, which are listed in Table 1 together with their respective flow type.

**Decoupling generators and flows.** By uncovering the role of the generator as a smoothing operator on vector fields, we suggest that the existence of a generator and the flow that particles follow can be decoupled. We deduce that it is possible to train a generator with score-based gradients which replace adversarial training (which we call a Score GAN); and that a GAN can be trained without a generator, using only the discriminator to synthesize samples (which we call a Discriminator Flow); cf. Table 1. We introduce these new models as proofs of concept, which we empirically assess to support the validity of our framework and illustrate the new perspectives it opens up in this active field of research.

Throughout the paper, we call out some specifics of the contributions of our framework as **Definitions** and **Findings** (*italicized* in the main text).

**Outline.** We begin with a discussion of particle models without generators in Section 2, covering Wasserstein gradient flows and score-based diffusion models. In Section 3 we discuss how generator training can be framed as a particle model with a generator, including GANs and Stein gradient flows. Section 4 then highlights how our framework allows us to decouple the generator and flow components of particle models, leading to the aforementioned new models, Score GANs and Discriminator Flows. Finally, we discuss the implications of our findings and conclude the paper in Section 5.

**Notations.** We consider the evolution of generated distributions $\rho_t$ over $\mathbb{R}^D$ w.r.t. time $t \in \mathbb{R}_+$, where $t$ is the inference time for particle models without generators, or the training time for generator training, respectively. This evolution until some finite or infinite end time $T \in \mathbb{R}_+ \cup \{+\infty\}$ then yields a final generated distribution $p_{\text{g}} = \rho_T$, which ideally approaches the data distribution $p_{\text{data}}$.

## 2 Particle Models without Generators: Non-interacting Particles

In this section we formally introduce the notion of generative particle models that do not use a generator, and present two standard instances: Wasserstein gradient flows and score-based diffusion models. They involve the manipulation of particles $x_t \sim \rho_t$ following a differential equation parameterized by some vector field. We characterize these models by noticing that their particles actually optimize, independently from one another, a loss that depends on the current particle distribution.

**Definition 1** (Particle Models, PMs). *PMs model particles $x_t \sim \rho_t$ starting from a prior $\rho_0 = \pi$:*

$$x_0 \sim \pi = \rho_0, \qquad\qquad \mathrm{d}x_t = \nabla h_{\rho_t}(x_t)\,\mathrm{d}t, \qquad\qquad (1)$$

*where $h_{\rho_t} \colon \mathbb{R}^D \to \mathbb{R}$ is a functional that depends on the current particle distribution $\rho_t$. Time $t$ corresponds to generation/inference time from $\rho_0$ to the final distribution $p_{\text{g}} = \rho_T$.*

**Finding 1.** *In PMs, the evolution of Equation (1) makes each generated particle $x_t$ individually follow a gradient ascent path on the objective $h_{\rho_t}(x_t)$.*

In prior works, the prior $\pi$ is conventionally chosen to be easy to sample from, such as a Gaussian. $h_{\rho_t}$ is usually defined in a theoretical manner so that the flow of Equation (1) conveys good convergence

properties for $\rho_t$ towards $p_{\text{data}}$ when $t \to T$. Because analytically computing $h_{\rho_t}$ is often intractable, it is empirically estimated and replaced by a neural network in practice.

## 2.1 Wasserstein Gradient Flows

A Wasserstein gradient flow is a generalization of gradient descent on a functional in the space of probability measures. More formally, it is an absolute continuous curve of probability distributions in a Wasserstein metric space $\mathcal{P}_2$ over $\mathbb{R}^D$ that satisfies a continuity equation (Santambrogio, 2017), and equivalently an evolution of particles $x_t \sim \rho_t$ under mild hypotheses (Jordan et al., 1998):

$$\partial_t \rho_t - \nabla \cdot \left( \rho_t \nabla_W \mathcal{F}(\rho_t) \right) = 0, \qquad \mathrm{d}x_t = -\nabla_W \mathcal{F}(\rho_t)(x_t) \, \mathrm{d}t, \qquad (2)$$

where $\mathcal{F} \colon \mathcal{P}_2 \to \mathbb{R}$ is the functional to minimize in the Wasserstein space. This definition involves the Wasserstein gradient of the functional $\mathcal{F}(\rho_t)$ – similar to the gradient of a functional defined over a Euclidean space – which for some functionals can be obtained in closed form by computing the first variation of the functional $\mathcal{F}$ (Santambrogio, 2017, Section 4.3):

$$\nabla_W \mathcal{F}(\rho_t) = \nabla \frac{\partial \mathcal{F}(\rho_t)}{\partial \rho_t} \colon \mathbb{R}^D \to \mathbb{R}^D. \qquad (3)$$

**Finding 2.** *Wasserstein gradient flows are PMs with* $h_{\rho_t} = -\dfrac{\partial \mathcal{F}(\rho_t)}{\partial \rho_t} + \text{cst}$ *and* $T = +\infty$.

The partial differential equation governing the particle evolution, as well as its convergence properties toward $p_{\text{data}}$, strongly depends on the functional $\mathcal{F}$. We detail the standard examples of the forward Kullback-Leibler (KL) divergence (Kullback & Leibler, 1951), $f$-divergences (Rényi, 1961), the squared Maximum Mean Discrepancy (MMD, Gretton et al., 2012; Arbel et al., 2019), and entropy regularization in Table 2. They can be additively combined for a variety of objectives $\mathcal{F}$. More

Table 2: Gradient flows for standard objectives $\mathcal{F}$.

| | Objective $\mathcal{F}(\rho)$ | $h_\rho$ |
|---|---|---|
| Forward KL | $\mathbb{E}_\rho \log \rho/p_{\text{data}}$ | $-\log \rho/p_{\text{data}}$ |
| $f$-divergence | $\mathbb{E}_{p_{\text{data}}} f\left(\rho/p_{\text{data}}\right)$ | $-f'\left(\rho/p_{\text{data}}\right)$ |
| Squared MMD w.r.t. kernel $k$ | $\displaystyle \mathop{\mathbb{E}}_{\substack{x,x' \sim \rho \\ y,y' \sim p_{\text{data}}}} \begin{bmatrix} k(x,x') \\ +k(y,y') \\ -2k(x,y) \end{bmatrix}$ | $\begin{array}{c} \mathbb{E}_{y \sim p_{\text{data}}}[k(y,\cdot)] \\ -\mathbb{E}_{x \sim \rho}[k(x,\cdot)] \end{array}$ |
| Entropy | $\mathbb{E}_\rho \log \rho$ | $-\log \rho$ |

examples exist in the literature (Liutkus et al., 2019; Mroueh et al., 2019; Glaser et al., 2021).

Several methods have been explored in the literature to solve Equation (2) in practice, either using input-convex neural networks (Mokrov et al., 2021; Alvarez-Melis et al., 2022) to discretize the continuous flow, or parameterizing $\nabla h_\rho$ by a neural network (Gao et al., 2019; Fan et al., 2022; Heng et al., 2023). In all cases these methods fit within the class of PMs as framed in Definition 1.

## 2.2 Score-Based Diffusion Models

Early score-based models (Noise Conditional Score Networks [NCSN], Song & Ermon, 2019) rely on Langevin dynamics as described in the following stochastic differential equation, converging towards $p_{\text{data}}$ when $t \to \infty$:

$$\mathrm{d}x_t = \nabla \log p_{\text{data}}(x_t) \, \mathrm{d}t + \sqrt{2} \, \mathrm{d}W_t. \qquad (4)$$

Several methods that use neural networks to estimate the score function of the data distribution $\nabla \log p_{\text{data}}$ (Hyvärinen, 2005), coupled with the use of Langevin dynamics, can work in practice even for high-dimensional distributions. Nonetheless, because of ill-definition and estimation issues of the score for discrete data on manifolds, a Gaussian perturbation of the data distribution is introduced to stabilize the dynamics. Thus, $p_{\text{data}}$ in Equation (4) is replaced by the distribution $p_{\text{data}}^\sigma$ of $x + \sigma\varepsilon$, where $x \sim p_{\text{data}}$ and $\varepsilon \sim \mathcal{N}(0, I_D)$. Denoting $\star$ as the convolution of a probability distribution $p$ by a kernel $k \colon \mathbb{R}^D \times \mathbb{R}^D \to \mathbb{R}$, we notice that $p_{\text{data}}^\sigma$ is actually the convolution of $p_{\text{data}}$ by a Gaussian kernel:

$$p_{\text{data}}^\sigma = p_{\text{data}} \star k_{\text{RBF}}^\sigma, \qquad p \star k \triangleq \int_x k(x, \cdot) \, \mathrm{d}p(x), \qquad k_{\text{RBF}}^\sigma(x, y) \triangleq \frac{1}{\sigma\sqrt{2\pi}} e^{-\frac{\|x-y\|_2^2}{2\sigma^2}}. \qquad (5)$$

NCSN then follows this equation, estimating the score with denoising score matching (Vincent, 2011) and repeating the process for a decreasing sequence of $\sigma$s:

$$\mathrm{d}x_t = \nabla \log[p_{\text{data}} \star k_{\text{RBF}}^\sigma](x_t) \, \mathrm{d}t + \sqrt{2} \, \mathrm{d}W_t. \qquad (6)$$

Building on Song et al. (2021), newer score models (Karras et al., 2022) make $\sigma$ a continuous function of time $\sigma(t)$, decreasing towards 0 in finite time to improve convergence. Many of these approaches share the following generation equation, corresponding to the reverse of a noising process for $p_{\text{data}}$ (Elucidating the Design Space of Diffusion-Based Generative Models [EDM], Karras et al., 2022):

$$\mathrm{d}x_t = 2\sigma'(t)\sigma(t)\nabla \log\left[p_{\text{data}} \star k_{\text{RBF}}^{\sigma(t)}\right](x_t)\,\mathrm{d}t + \sqrt{2\sigma'(t)\sigma(t)}\,\mathrm{d}W_t, \tag{7}$$

where $\sigma'(t)$ is the derivative of $\sigma(t)$. These approaches admit an equivalent deterministic flow yielding the same probability path $\rho_t$:

$$\mathrm{d}x_t = \sigma'(t)\sigma(t)\nabla \log\left[p_{\text{data}} \star k_{\text{RBF}}^{\sigma(t)}\right](x_t)\,\mathrm{d}t. \tag{8}$$

However, we notice that this only holds under the implicit assumption that Equation (7) perfectly reverses the initial noising process, i.e., $\rho_t = p_{\text{data}} \star k_{\text{RBF}}^{\sigma(t)}$. This assumption can be broken in practice when the score is not well estimated or for coarse time discretization. In the general case for both NCSN and EDM, by using the Fokker-Planck equation (Jordan et al., 1998), which allows us to substitute the stochastic component $\mathrm{d}W_t$ by the negative of the score of the generated distribution, we obtain, respectively, the following equivalent exact probability flows:

$$\frac{\mathrm{d}x_t}{\mathrm{d}t} = \nabla \log\left[\frac{p_{\text{data}} \star k_{\text{RBF}}^{\sigma}}{\rho_t}\right](x_t), \qquad \frac{\mathrm{d}x_t}{\mathrm{d}t} = \sigma'(t)\sigma(t)\nabla \log\left[\frac{1}{\rho_t}\left(p_{\text{data}} \star k_{\text{RBF}}^{\sigma(t)}\right)^2\right](x_t). \tag{9}$$

**Finding 3.** *Score-based diffusion models are PMs:*

- *NCSN (Song & Ermon, 2019) with $h_{\rho_t} = \log\left[p_{\text{data}} \star k_{\text{RBF}}^{\sigma}\right] - \log \rho_t$ and $T = +\infty$;*

- *EDM (Karras et al., 2022) with $h_{\rho_t} = \sigma'(t)\sigma(t)\left(2\log\left[p_{\text{data}} \star k_{\text{RBF}}^{\sigma(t)}\right] - \log \rho_t\right)$ and $T < +\infty$.*

NCSN actually implements through Langevin dynamics a forward KL gradient flow, which is common knowledge in the related literature (Jordan et al., 1998; Yi et al., 2023).

## 3 Particle Models with Generators: Training of Interacting Particles

In the previous section we presented a framework for particle models (PMs) that, in the absence of a generator, individually manipulate particles in the data space, and optimize a distribution-dependent objective $h_{\rho_t}$ via a differential equation. In this section we frame generator training as a generalization of PMs involving direct interaction between particles. We show, supported by prior literature, that our framework applies to the case of GANs and Stein gradient flows.

### 3.1 Generator Training as a Modified Particle Model

We begin with the training of a neural generator $g_\theta \colon \mathbb{R}^d \to \mathbb{R}^D$ parameterized by $\theta$. Associated with a prior distribution on its latent space $p_z$, $g_\theta$ produces a generated distribution $p_\theta$ as the pushforward of $p_z$ through $g_\theta$: $p_\theta = g_\theta \sharp p_z$, by which we seek to imitate $p_{\text{data}}$. Unlike PMs which progressively construct synthesized samples directly in the data space $\mathbb{R}^D$, generators enable models like GANs to generate samples starting from a different latent space. When $d < D$, this latent space allows the resulting distribution to be naturally embedded into a lower-dimensional manifold, thereby integrating the manifold hypothesis (Bengio et al., 2013). The parameters $\theta$ evolve during training, making the generated distribution move accordingly: $\rho_t = p_{\theta_t}$.

We characterized in Section 2, Finding 1, PMs as models that make free generated particles optimize an objective $h_\rho \colon \mathbb{R}^D \to \mathbb{R}$ that conveys desirable convergence properties. We leverage this observation to show that generator training can be framed as a PM as well. We see that generators involve generated particles $x_t \sim \rho_t$ as generator outputs $x_t = g_{\theta_t}(z)$, with $z \sim p_z$, which move during training. We proceed by making the generator optimize the same objective $h_{\rho_t}$ as in PMs, that is, the generator parameters are trained to minimize at each optimization step:

$$\mathcal{L}_{\text{gen}}(\theta) = -\mathbb{E}_{z \sim p_z}\left[h_{\rho_t}\left(g_\theta(z)\right)\right]. \tag{10}$$

In the optimization of Equation (10), we intentionally ignore the dependency of $\rho$ on $\theta$, i.e. in practice $\rho = \text{StopGradient}(g_\theta \sharp p_z)$. This allows us to mimic PMs where generated particles $x_t$s optimize the objective $h_{\rho_t}$, without taking into account that $\rho_t$ is actually a mixture of all the $x_t$s.

By optimizing $\theta_t$ via gradient descent for the loss of Equation (10) with learning rate $\eta$, idealized in the continuous training time setting, we obtain using the chain rule:

$$\frac{\mathrm{d}\theta_t}{\mathrm{d}t} = \eta \nabla_{\theta_t} \mathbb{E}_{z \sim p_z}\Big[h_{\rho_t}\big(g_{\theta_t}(z)\big)\Big] = \eta \mathbb{E}_{z \sim p_z}\Big[\nabla_{\theta_t} g_{\theta_t}(z) \nabla h_{\rho_t}\big(g_{\theta_t}(z)\big)\Big]. \tag{11}$$

As a consequence, using the chain rule again, each generated particle $x_t = g_{\theta_t}(z) \sim \rho_t$ evolves as:

$$\frac{\mathrm{d}g_{\theta_t}(z)}{\mathrm{d}t} = \nabla_{\theta_t} g_{\theta_t}(z)^\top \frac{\mathrm{d}\theta_t}{\mathrm{d}t} = \eta \mathbb{E}_{z' \sim p_z}\Big[k_{g_{\theta_t}}\big(z, z'\big) \nabla h_{\rho_t}\big(g_{\theta_t}(z')\big)\Big], \tag{12}$$

where $k_{g_\theta}\colon z, z' \mapsto \nabla_{\theta_t} g_{\theta_t}(z)^\top \nabla_{\theta_t} g_{\theta_t}(z')$ is the matrix Neural Tangent Kernel (NTK, Jacot et al., 2018) of the generator. Equation (12) describes the dynamics of the generated particles as a modified version of Equation (1) for PMs. We formalize this as follows.

**Definition 2** (Interacting Particle Models, Int-PMs). *Int-PMs model particles resulting from the pushforward $g_{\theta_t} \sharp p_z$ of a generator $g_{\theta_t}$ applied to a prior $p_z$, with the following training dynamics:*

$$\mathrm{d}g_{\theta_t}(z) = \eta\big[\mathcal{A}_{\theta_t}(z)\big]\big(\nabla h_{\rho_t}\big)\,\mathrm{d}t, \tag{13}$$

*where $h_{\rho_t}\colon \mathbb{R}^D \to \mathbb{R}$ is a functional that depends on the current distribution $\rho_t$, time $t$ is training time, and $\mathcal{A}_{\theta_t}(z)$ is a linear operator operating on vector fields (Sriperumbudur et al., 2010), defined as:*

$$\big[\mathcal{A}_{\theta_t}(z)\big](V) \triangleq \mathbb{E}_{z' \sim p_z}\Big[k_{g_{\theta_t}}\big(z, z'\big) V\big(g_{\theta_t}(z')\big)\Big], \quad k_{g_{\theta_t}}\big(z, z'\big) \triangleq \nabla_{\theta_t} g_{\theta_t}(z)^\top \nabla_{\theta_t} g_{\theta_t}(z'). \tag{14}$$

Similarly to PMs, the vector field $\nabla h_{\rho_t}$ in Int-PMs indicates which direction each generated particle will follow to get closer to the data distribution. However, $\mathcal{A}_{\theta_t}$ smooths this gradient field using the generator's NTK, and generated particles thus interact with each other. Indeed, moving one particle makes its neighbors move accordingly because of their underlying parameterization by the generator. Notably, Int-PMs generalize PMs: in the degenerate case where $k(z, z') = \delta_{z-z'} I_D$, with $\delta$ the Dirac delta function centered on $0$, i.e., when particles can move freely with a sufficiently powerful generator, the effect of parameterization disappears with $\big[\mathcal{A}_{\theta_t}(z)\big](V) = V\big(g_\theta(z)\big)$, and therefore Equation (13) reduces to Equation (1).

**Finding 4.** *Int-PMs generalize PMs. Each Int-PM is therefore defined by two components: the objective function $h_{\rho_t}$ and the choice of generator architecture $g_\theta$.*

We will show in the remainder of this section that Int-PMs encompass both GANs and Stein gradient flows, borrowing Franceschi et al. (2022)'s results which the previous reasoning generalizes.

## 3.2 GANs as Interacting Particle Models

In GANs, each generator $g_\theta$ is accompanied by a discriminator $f_\rho$ that depends on the generated distribution. $f_\rho$ is optimized as a neural network via gradient ascent (GA) to maximize an objective of the following form:

$$f_\rho = \text{GA}_f\Big\{\mathcal{L}_\mathrm{d}(f; \rho, p_\text{data}) \triangleq \mathbb{E}_\rho[a \circ f] - \mathbb{E}_{p_\text{data}}[b \circ f] + \mathcal{R}(f; \rho, p_\text{data})\Big\}, \tag{15}$$

for some functions $a, b\colon \mathbb{R} \to \mathbb{R}$ (e.g. for the WGAN of Arjovsky et al. (2017), $a = b = \text{id}$) and regularization $\mathcal{R}$ (e.g., the gradient penalty of Gulrajani et al. (2017)). In this work, we remain oblivious to how the discriminator is trained in practice as a single network alongside the generator. Nonetheless, we note that this GA is usually stopped early and not run until convergence, because the discriminator is trained only for a few steps between generator updates.

This discriminator is then used to train the generator, as usually framed in a min-max optimization setting. However, several works (Metz et al., 2017; Franceschi et al., 2022; Yi et al., 2023) showed that generator optimization deviates from min-max optimization, because alternating updates between the generator and the discriminator make the generator minimize a loss function of the form (Franceschi et al., 2022):

$$\mathcal{L}_\text{GAN}(g_\theta) = \mathbb{E}_{z \sim p_z}\Big[\big(c \circ f_\rho\big)\big(g_\theta(z)\big)\Big], \tag{16}$$

for some $c\colon \mathbb{R} \to \mathbb{R}$ (e.g., for WGAN, $c = \text{id}$). Using Definition 2, we deduce the following.

**Finding 5.** *GANs are Int-PMs with $h_\rho = -c \circ f_\rho$, where $f_\rho$ is the current discriminator.*

Under some assumptions on the outcome of discriminator training in Equation (15), the resulting $\nabla h_\rho$ for GANs has been proven to implement a Wasserstein gradient $-\nabla_W \mathcal{F}(\rho)$. Two notable examples are (see also Section 2.1): $f$-divergence GANs (Nowozin et al., 2016), which are linked to the forward KL divergence gradient flow (Yi et al., 2023) and therefore to diffusion models; and Integral Probability Metrics (IPM) GANs, which are linked to the squared MMD gradient flow w.r.t. the NTK of the discriminator (Franceschi et al., 2022). However, these links have been made under strong simplifying assumptions, and the GAN formulation as an Int-PM in this paper is far more general.

**Finding 6.** *$f$-divergence GANs as Int-PMs generalize the forward KL gradient flow and Langevin diffusion models, and IPM GANs generalize MMD gradient flows.*

### 3.3 Stein Gradient Flows as Int-PMs

Int-PMs as framed in Definition 2 are similar to Stein gradient flows (Liu & Wang, 2016; Liu, 2017; Duncan et al., 2023). The latter are a generalization of Wasserstein gradient flows in another geometry shaped by a matrix kernel $k \colon \mathbb{R}^D \times \mathbb{R}^D \to \mathbb{R}^{D \times D}$ defined in the data space:

$$\mathrm{d}x_t = -\mathbb{E}_{x_t' \sim \rho_t}\Big[ k\big(x_t, x_t'\big) \nabla_W \mathcal{F}(\rho_t)\big(x_t'\big) \Big]\, \mathrm{d}t. \tag{17}$$

Prior works showed strong links between such flows and GAN optimization, which help us to see that GANs are Int-PMs. In the following, we will see that Stein Gradient flows serve as an example of a functional generator-based counterpart of a generator-less PM.

We begin by generalizing the reasonings of Chu et al. (2020), Durr et al. (2022) and Franceschi et al. (2022), which initially applied only to the case of GANs. We assume that for Int-PMs in Equations (13) and (14), the generator's NTK is constant throughout training, i.e. $k_{g_{\theta_t}} = k_g$, like for many networks with infinite width (Jacot et al., 2018; Liu et al., 2020). Then, when $\nabla h_\rho = -\nabla_W \mathcal{F}(\rho_t)$, e.g., for gradient flows as in Finding 2 or for GANs following Finding 5, we obtain an equation similar to Equation (17):

$$\frac{\mathrm{d}g_{\theta_t}(z)}{\mathrm{d}t} = -\eta \mathbb{E}_{z' \sim p_z}\Big[ k_g\big(z, z'\big) \nabla_W \mathcal{F}(\rho_t)\big(g_{\theta_t}(z')\big) \Big]. \tag{18}$$

This is a special case of Equation (17) with an invertible generator (Chu et al., 2020). However, in the general case, the kernel $k_g \colon \mathbb{R}^d \times \mathbb{R}^d \to \mathbb{R}^{D \times D}$ acts on the latent space, which makes Equation (18) define a generalized latent-driven Stein flow that was uncovered by Franceschi et al. (2022).

**Finding 7.** *Stein gradient flows are Int-PMs with an invertible generator in the NTK regime and $h_{\rho_t} = -\frac{\partial \mathcal{F}(\rho_t)}{\partial \rho_t} + \mathrm{cst}$. They can be generalized to non-invertible generators with Equation (18).*

## 4 Decoupling the Generator and the Flow in Particle Models

We saw in the last section that Int-PMs, such as GANs, generalize PMs, such as diffusion and Wasserstein gradient flows, by applying their generative equations to generator training. This implies that many generative models can be defined by their PM flow, which allows an optional generator to be trained. From Section 3.3, this is the case for gradient flows, which can either define a PM or an Int-PM with the same $\nabla h$ as a Wasserstein gradient. As per Finding 6, this also holds for GANs, which share a common particle movement with gradient flows and diffusion models.

Consequently, our framework suggests that a functional $h_\rho$ used in an Int-PM may equivalently be used in a PM, and vice versa. This leads us to formulate the following claim.

**Claim 1.** *A generator can be trained using the gradient flow of a score-based diffusion model instead of adversarial training, and it is possible to remove the generator in a GAN by synthesizing samples with the discriminator only.*

We confirm this hypothesis in this section by introducing corresponding new hybrid models, which we call respectively Score GANs and Discriminator Flows (see Table 1), and by empirically demonstrating their viability. Note that we introduce these models as proofs of concepts of the applications of our framework. Since they challenge many assumptions and standard practices of generative modeling, they do not benefit from the same wealth of accumulated knowledge that classic models have access to, and thus are harder to tune than standard models.

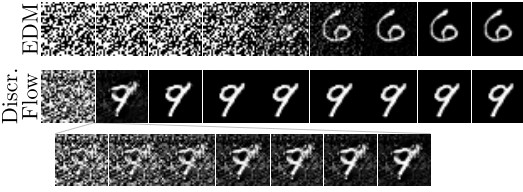

Figure 1: From left to right, generation process on MNIST of EDM and Discriminator Flow for every 8 evaluations of $\nabla h_{\rho_t}$. The last row shows the first 7 steps of Discriminator Flow.

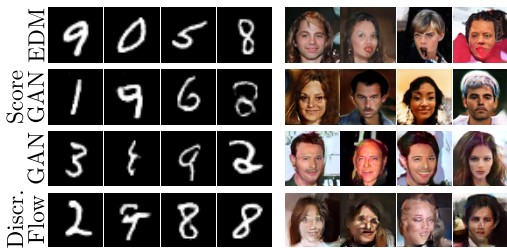

Figure 2: Uncurated samples of studied models on CelebA and MNIST.

---

**Algorithm 1:** Training iteration of Score GANs; all operations can be performed in parallel for batching. See Appendix B for details on the practical implementation of lines 3 and 5.

**Input:** Noise distribution $p_\sigma$, number of intermediate score training steps $K$, learning rates $\lambda, \eta \in \mathbb{R}_+$, previous generator $g_\theta \colon \mathbb{R}^D \to \mathbb{R}$, previous $\rho$ score model $s_\phi^\rho \colon \mathbb{R}^D \times \mathbb{R} \to \mathbb{R}^D$, pretrained $p_{\text{data}}$ score model $s_\psi^{p_{\text{data}}} \colon \mathbb{R}^D \times \mathbb{R} \to \mathbb{R}^D$.

**Output:** Updated generator $g_\theta$ and $\rho$ score model $s_\phi^\rho$.

1 **for** $k = 1$ **to** $K$ **do**              // Updates of $s^\rho$ by denoising score matching

2  $\quad z \sim p_z, x = g_\theta(z), \sigma \sim p_\sigma, x^\sigma \sim \mathcal{N}\big(x, \sigma^2 I_D\big)$;

3  $\quad \phi \leftarrow \phi - \lambda \nabla_\phi \Big\| s_\phi^\rho(x^\sigma, \sigma) + \frac{x^\sigma - x}{\sigma^2} \Big\|_2^2$

   // Score matching with generator, Equations (11) and (19)

4 $z \sim p_z, \sigma \sim p_\sigma, \varepsilon \sim \mathcal{N}(0, I_D)$;

5 $\theta \leftarrow \theta + \eta \cdot \nabla_\theta g_\theta(z)^\top \Big( s_\psi^{p_{\text{data}}}\big(g_\theta(z) + \sigma\varepsilon, \sigma\big) - s_\phi^\rho\big(g_\theta(z) + \sigma\varepsilon, \sigma\big) \Big)$

---

**Experimental setting.** We conduct experiments on the unconditional generation task for two standard datasets composed of images: MNIST (LeCun et al., 1998) and $64 \times 64$ CelebA (Liu et al., 2015). We consider two reference baselines, EDM (the score-based diffusion model of Karras et al. (2022)) and GANs, and use the Fréchet Inception Distance (FID, Heusel et al., 2017) to test generative performance

Table 3: Test FID of studied models.

| Dataset | PMs (no generator) | | Int-PMs (generator) | |
|---|---|---|---|---|
| | EDM | Discr. Flow | GAN | Score GAN |
| MNIST | 3 | 4 | 3 | 15 |
| CelebA | 10 | 41 | 19 | 35 |

in Table 3. Training details are given in Appendix D; our open-source code is available at `https://github.com/White-Link/gpm`. We refer to Appendix C and the code for more experimental results and samples for each baseline.

### 4.1 Training Generators with Score-Based Diffusion: Score GANs

We propose training a generator with the score-based diffusion flow of NCSN, Equation (9), left. This involves applying Equation (13) with $h_{\rho_t} = \log\big[p_{\text{data}} \star k_{\text{RBF}}^\sigma\big] - \log \rho_t$, where $t$ represents the training time of the generator. To do so, we directly use the generator weight update formula of Equation (11), as this avoids the problem of estimating $h_{\rho_t}$ and only requires its gradient $\nabla h_{\rho_t} = \nabla \log\big[p_{\text{data}} \star k_{\text{RBF}}^\sigma\big] - \nabla \log \rho_t$. Composed of the scores of, respectively, the noised data distribution and the generated distribution, $\nabla h_{\rho_t}$ can be efficiently estimated via score matching techniques.

In practice, we use a score network $s_\psi^{p_{\text{data}}}$ pretrained with the latest denoising score matching techniques (Karras et al., 2022) to estimate the static term $\nabla \log\big[p_{\text{data}} \star k_{\text{RBF}}^\sigma\big]$. Moreover, as $\nabla \log \rho_t$ is dynamic and needs to be continuously estimated, we leverage GAN discriminator practices and train a network $s_\phi^\rho$ by alternating with generator updates to estimate this score.

However, our proposed solution remains impractical for two reasons. First, since the dynamics would match $p_{\text{data}} \star k_{\text{RBF}}^\sigma$ and $\rho_t$, we would need to schedule $\sigma$s during training, similar to what Song & Ermon (2019) do during inference. Second, while $\nabla \log \rho_t$ can be estimated using sliced score

**Algorithm 2:** Training iteration of Discr. Flows. Cf. batching and discretization in Appendix B.

**Input:** Initial distribution $\pi = \rho_0$, gradient strength $\eta \in \mathbb{R}_+$, learning rate $\lambda \in \mathbb{R}_+$, previous discriminator $f_\phi \colon \mathbb{R}^D \times \mathbb{R} \to \mathbb{R}$.
**Output:** Updated discriminator $f_\phi$.

1 $x \sim p_{\text{data}}, x_0 \sim \pi, t \sim \mathcal{U}(0,1)$;     // Initialization, random sampling time

2 $x_t \leftarrow x_0 - \eta \int_0^t \nabla_{x_s} \left[ (c \circ f_\phi)(x_s, s) \right] \mathrm{d}s$;   // Partial generation, Equation (20)

3 $\phi \leftarrow \phi + \lambda \nabla_\phi \left\{ \mathcal{L}_{\mathrm{d}}\big( f_\phi(\cdot, t); \delta_{x_t}, \delta_x \big) \right\}$;   // Train $f_\phi$ at time $t$, cf. Equation (15)

matching (Song et al., 2020), this approach is less performant than denoising score matching and leads to estimation issues when $\rho_t$ lies on a manifold (Song & Ermon, 2019), as in our case with a pushforward generator. Both of these problems can be solved by instead matching $p_{\text{data}} \star k_{\text{RBF}}^\sigma$ and $\rho_t \star k_{\text{RBF}}^\sigma$ for a range of $\sigma \sim p_\sigma$, using Equations (12) and (13):

$$h_{\rho_t} = \log[p_{\text{data}} \star k_{\text{RBF}}^\sigma] - \log[\rho_t \star k_{\text{RBF}}^\sigma]. \tag{19}$$

This approach allows us to leverage denoising score matching to train $s_\phi^\rho$, and to avoid scheduling $\sigma$s by instead sampling them during training and noising the generated distribution with the chosen noise levels. Overall, we obtain the algorithm for Score GANs described in Algorithm 1.

**Finding 8.** *Score GANs are Int-PMs with* $h_\rho = \mathbb{E}_{\sigma \sim p_\sigma} \left[ \log \left[ \frac{p_{\text{data}} \star k_{\text{RBF}}^\sigma}{\rho \star k_{\text{RBF}}^\sigma} \right] \right]$ *and* $T = +\infty$.

Like in GANs, the generated score network $s_\phi^\rho$ is trained for a small number of steps $K$ in between two generator updates. Accordingly, as is the case for a discriminator, $K$ is an important parameter for Score GANs. We study its impact of the method's performance in Appendix C.4, where we ensure that a small $K$ suffices to accurately estimate the score of the generated distribution.

## 4.2 Removing the Generator from GANs: Discriminator Flows

Based on Findings 4 and 5, we see that removing the generator from GANs to make them PMs, as in Definition 1, simply requires that we define $h_{\rho_t} = -c \circ f_{\rho_t}$, where $f_{\rho_t}$ is the discriminator between $\rho_t$ and $p_{\text{data}}$ at sampling time $t$ in Equation (1):

$$\mathrm{d}x_t = -\nabla\big(c \circ f_{\rho_t}\big)(x_t)\, \mathrm{d}t. \tag{20}$$

This makes Equation (20) the equivalent of GAN training, but as a PM without a generator. In other words, Discriminator Flows make individual particles follow the gradient of the generator loss of Equation (16), defined through the discriminator, without the generator smoothing of Equation (13).

**Finding 9.** *Discriminator Flows are PMs with* $h_\rho = -c \circ f_{\rho_t}$ *and* $T = +\infty$.

Such a model would a priori require us to successively train a neural discriminator $f_{\phi_t} \equiv f_{\rho_t}$ per time step $t$. This results, however, in a prohibitively slow and heavy training procedure. As a more scalable alternative, we train a single time-dependent neural discriminator $f_\phi \colon \mathbb{R}^D \times \mathbb{R} \to \mathbb{R}$, that takes as input both a sample $x_t \sim \rho_t$ and its corresponding time $t \in \mathbb{R}$, on all time steps at once. Each $x_t \sim \rho_t$ must then be computed both in training and inference from $x_0 \sim \rho_0 = \pi$ using Equation (20). This results in the training procedure of Algorithm 2 for Discriminator Flows. For practical convenience, we restrict, without loss of generality, $t \in [0, 1]$.

Compared to diffusion models, for which the score can be freely estimated at each $t$ because $\rho_t$ is assumed to equal $p_{\text{data}} \star k_{\text{RBF}}^{\sigma(t)}$, it is more challenging to elucidate the particle movement of Equation (20). Prior studies on GAN optimization help answer this problem under simplifying assumptions. Indeed, if $a$, $b$, and $c$ (cf. Section 3.2) are chosen to implement an $f$-divergence GAN loss, then Discriminator Flows will implement a forward KL divergence gradient flow (Yi et al., 2023) If they instead correspond to an IPM GAN loss, then Discriminator Flows will implement a squared MMD gradient flow (Franceschi et al., 2022).

In the general case, the generation process of Discriminator Flows is not known in advance. In fact, we must simulate the entire process during training, making each training iteration slower. However,

our approach has the advantage of generalizing the diffusion process and allowing the discriminator to learn another path from the initial distribution $\rho_0 = \pi$ to $p_{\mathrm{data}}$. We believe that, when properly tuned, Discriminator Flows could provide faster sampling times than diffusion models because they can learn shorter paths towards $p_{\mathrm{data}}$.

## 4.3 Experimental results

We see uncurated samples from our proposed Score GAN and Discriminator Flow models in Figure 2, as well as the baseline EDM and GAN models. In Table 3 we report the FID scores for all models. We observe that both of our introduced models produce reasonable results, which experimentally confirms our initial claim. Nonetheless, these models exhibit worse (higher) FID scores than our baselines, although Discriminator Flows provide good performance on the MNIST dataset. We attribute these performance results to the fact that our proposed models are novel and do not benefit from the accumulated knowledge regarding training best practices that the standard models possess.

Interestingly, the proposed models exhibit typical properties of generator-free and generator-based models. Since Score GANs use a generator, as in GANs, only one function evaluation is required to draw a sample, making it orders of magnitude faster to sample from than EDM. Like GANs, Score GANs may produce mode collapse and benefit from smooth latent space interpolations, as shown in Appendices C.2 and C.3. Of course, Discriminator Flows are slower at both training and inference time than such generator-based models, but this comes with the additional flexibility of operating directly in the data space (Voleti et al., 2022; Couairon et al., 2023). As a consequence, like diffusion models, the latent space interpolation (i.e., interpolating the initial noise $x_0 \sim \pi$ in Equation (1)) capabilities of Discriminator Flows remain below those of generator-based models – cf. Appendix C.3.

However, as shown in Figure 1 and in the numerical results of Appendix C.5, Discriminator Flows have the expected advantage of converging to $p_{\mathrm{data}}$ faster than the state-of-the-art diffusion model EDM. In particular, we notice that images generated by Discriminator Flows are well-formed early in the generation process, hinting at potential temporal cost reduction by stopping the process early. Unfortunately, this has not yet yielded a better time efficiency than EDM; we nevertheless believe it can be achieved with additional architectural and model tuning of Discriminator Flows. We further discuss the time efficiency of our introduced methods in Appendices A.4, C.5 and D.3.

## 4.4 Relationship with Prior Work

**Score GANs.**  Conceptually, Score GANs implement for each $\sigma$ a forward KL gradient flow, similarly to Yi et al. (2023), who theoretically proved that some GAN models approximate such a flow without noising, under optimality assumptions on the discriminator. However, Score GANs differ from traditional GANs as they do not involve a discriminator, but rather split the flow $\nabla h_{\rho_t}$ into two parts: one part that can be estimated before generator training as it depends only on the data distribution, and another part that must be continuously estimated during training as it depends on the generated distribution. While $h_{\rho_t}$ could be estimated by adding noise to the inputs of a discriminator (Wang et al., 2022), Score GANs instead only need to estimate the score of the generated distribution, which is no longer adversarial.

**Discriminator Flows.**  As a general concept, Discriminator Flows provide an encompassing framework that helps to reveal the connections of various approaches to GAN training.

Recently, Heng et al. (2023) introduced deep generative Wasserstein gradient flows (DGGF), a method that relies on $f$-divergence gradient flows approximated by estimating the ratio $\rho_t/p_{\mathrm{data}}$ with a neural network. Examined under our framework, DGGF's training objectives correspond to that of a discriminator in $f$-divergence GANs, allowing us to frame DGGF as a special case of Discriminator Flows. Nonetheless, we stress that Discriminator Flows have a larger scope than DGGF since we can handle all types of GAN objectives; all our experiments were performed with WGAN objectives. Moreover, unlike DGGF, which in practice removes the time dependency in its estimation without a theoretical justification, our method does handle time as input to the discriminator.

Discriminator Flows also relate to, and generalize, methods that finetune GAN outputs with gradient flows (Tanaka, 2019; Che et al., 2020; Ansari et al., 2021). The latter use discriminator gradients to approximate such flows, making them naturally expressible as Discriminator Flows in our framework.

In practice, they only apply their sampling procedures in the latent space of the generator, as applying them in pixel space leads to artifacts. We resolve this issue with a principled training procedure for the discriminator conditioned on sampling time.

We note that previous works, such as Xiao et al. (2022) and Jolicoeur-Martineau et al. (2021), also combine score-based models and GANs. Xiao et al. (2022) train several GANs to successively denoise an image, mimicking a reverse diffusion process. Jolicoeur-Martineau et al. (2021) augment the denoising objective of score-based diffusion models with an adversarial objective to improve the denoising image quality. However, while these works do draw links between both models, they do not aim to unify score-based models and GANs.

Finally, we provide intuition on the sampling efficiency of Discriminator Flows observed in Figure 1. We observe that diffusion models smooth the data distribution with a Gaussian kernel by Equation (5), while discriminators were shown to smooth the data distribution with their NTK (Franceschi et al., 2022). This observation brings diffusion models and GANs closer to each other, while explaining the fast convergence speed of Discriminator Flows thanks to the properties of NTKs for generative modeling. Indeed, Franceschi et al. (2022) presented empirical evidence that using NTKs of standard discriminators as kernels in squared MMD gradient flows (which some GAN models implement, cf. Finding 6), instead of Gaussian kernels, accelerates the convergence of Equation (1) towards $p_{\text{data}}$ by several orders of magnitude. More generally, we hypothesize that it is the natural effectiveness of neural networks in high dimensions that endows Discriminator Flows with a faster convergence speed than diffusion models, for which the particles' paths are chosen explicitly.

## 5    Conclusion

In this paper we have unified score-based diffusion models, GANs, and gradient flows under a single framework based on particle models which can be complemented with a generator. Since this framework unifies models that have been customarily opposed in the literature, this work paves the way for new perspectives in generative modeling. As an example of potential applications, we have shown that our framework naturally leads to two novel generative models: a generator that follows score-based gradients, and a generator-free GAN that uses a discriminator-guided generation process.

Of course, generator-less and generator-based models each retain their unique attributes. On the one hand, generator training provides a simple and efficient sampling procedure and endows the generative model with a low-dimensional structured latent space, at the cost of potential instability and mode collapse. On the other hand, generator-less models, despite their slow sampling, may be easier to train, since the generator component has been removed from their flow, and are more flexible as they rely on a continuous-time process directly defined in the data space. We believe that our framework, by revealing the close relationship between these models, can help them to improve upon one another, or can even help create other new hybrid models.

Beyond potential applications, our study could be enhanced and expanded in many ways for future work. On the theoretical side, we would like to tackle the challenging task (Hsieh et al., 2021) of taking into account the fact that the discriminator in GANs is actually continuously trained with the generator. It would also be interesting to generalize our framework to second-order and stochastic particle movement in generator-based models, and to study the impact of discretization on the studied differential equations, as hinted in Appendices A.1 to A.3. On the practical side, while we have proposed new models that function reasonably well, we would be interested in refining them further for state-of-the-art generative performance. Furthermore, Score GANs could serve as a distillation method for score-based diffusion models (Salimans & Ho, 2022), while Discriminator Flows could outperform diffusion models for generation efficiency.

## Acknowledgments and Disclosure of Funding

We would like to thank Lorenzo Croissant and Ugo Tanielian for helpful discussions and comments on this paper, as well as Edouard Delasalles for inspiring the architecture of our code.

This work was granted access to the HPC / AI resources of IDRIS under the allocation 2023-AD011013503R1 made by GENCI (Grand Equipement National de Calcul Intensif). Emmanuel de Bézenac is financially supported by the ETH Foundations of Data Science.

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

# A  Further Discussion

In this section we develop a number of elements of our framework and models to provide more context and perspective on our contributions.

## A.1  Extension to Stochastic Particle Models

We focus in the main paper on deterministic particle movement. However, some PMs like diffusion models also have equivalent stochastic particle movements as described in Equations (4) and (7). We provide some evidence in this section that stochastic particle movement can also be integrated to Int-PMs, thereby strengthening the generality of our approach.

Generally, using the Fokker-Planck equation, the following equation shares the same probability path as both Equations (7) and (9), for any $\alpha \in [0, 1]$:

$$\mathrm{d}x_t = 2\sigma'(t)\sigma(t)\nabla \log\Big[p_{\mathrm{data}} \star k_{\mathrm{RBF}}^{\sigma(t)}\Big](x_t)\,\mathrm{d}t - \alpha\sigma'(t)\sigma(t)\nabla \log \rho_t(x_t)\,\mathrm{d}t + \sqrt{2\alpha\sigma'(t)\sigma(t)}\,\mathrm{d}W_t. \tag{21}$$

This corresponds to an interpolation between Equations (7) and (9) that trades Brownian noise with its deterministic equivalent.

This stochastic component can then be integrated into the formulation of interacting particle models, via Equation (13). This latter equation takes the directions followed by the particles in the particle model, and transforms them via the operator $\mathcal{A}_{\theta_t}(z)$. Since this operator is linear, it is possible to integrate a stochastic component into the equation (Klebaner, 2012), allowing us to take into account stochastic particle models:

$$\mathrm{d}g_{\theta_t}(z) = \big[\mathcal{A}_{\theta_t}(z)\big]\big(\nabla h_{\rho_t}\,\mathrm{d}t + \gamma(t)\,\mathrm{d}W_t\big), \tag{22}$$

where $\gamma(t)$ is a scalar function of time.

This makes it possible to integrate the stochastic component of diffusion models into Score GANs by interpolating between Gaussian noise and the score of the generated distribution in step 5 of Algorithm 1, similarly to the previous equation using $\alpha$. Nonetheless, this comes with no guarantee on experimental performance, as we found in preliminary experiments that adding such a stochastic component is often detrimental to the resulting FID. Indeed, to succeed, the chosen gradient vector field to follow with the generator must be compatible with the generator architecture (i.e., compatible with the generator preconditioning $\mathcal{A}_{\theta_t}(z)$), which may not be the case with white noise of high variance.

## A.2  Discretizing Continuous-Time Equations

Studying how discretizing the considered continuous-time phenomena could affect our formulations is an interesting perspective for future work. We initiate a discussion on this topic below.

Choosing the best discretization method for diffusion models is challenging (Karras et al., 2022); since this depends on the chosen $\sigma(t)$, its efficiency is assessed w.r.t. the number of score queries instead of the number of discretization steps like in numerical methods, and the final purpose of discretization (generating realistic data) differs from its initial purpose (approximating a solution to a differential equation). Therefore, standard approaches like EDM rely on empirical discretization grids and custom solvers, tailored to the generation task. Our framework, by identifying the true probability flow in Equation (9), may help diffusion models cope with discretization errors through the score of the generated distribution.

The previous discussion, however, only holds for score-based diffusion models, for which the probability path is known in advance. This is not possible for other particle models, and studying the convergence properties of their discretizations, like Arbel et al. (2019) do for MMD, is non-trivial.

Adding a generator is an alternative way to solve the underlying particle model differential equation. By generalizing the parallel between Wasserstein and Stein gradient flows in Section 3.3, generators can be seen in our framework as a preconditioning over the particle model differential equation via the linear operator $\mathcal{A}_{\theta_t}(z)$ in Equation (13). A well-chosen architecture, adapted to the particle flow $\nabla h$, may speed up and simplify the dynamics towards the data distribution.

## A.3 Second-Order Methods and Adam

The framework we introduce relies on first-order solvers for PMs and first-order optimization for Int-PMs. Yet, in practice, we use Adam (Kingma & Ba, 2015), a second-order momentum-based optimizer, to train generators – cf. Appendix D.2. Theoretically, it is entirely possible to formulate continuous-time equations for Adam, paving the way for a generalization of our results to a second-order setting.

By fixing the values of $(\beta_1, \beta_2)$ and allowing the time step to approach zero, we can recover the continuous version of SignSGD. A relevant example of SignSGD's study within a non-convex context was presented by Bernstein et al. (2018). Furthermore, exploring the scenario where non-interaction is present (when $\mathcal{A}_{\theta_t}$ disappears from Equation (13) as discussed in Section 3.1) reveals a particle gradient flow with a renormalized gradient. This yields intriguing connections with continuous acceleration, as demonstrated by Wibisono & Wilson (2015). We consider this avenue to be a promising direction for future investigation.

## A.4 Time Efficiency of Score GAN and Discriminator Flow

The design of Score GAN and Discriminator Flow induces computational constraints that make each of their training iterations slower than those of baseline models. Score GAN requires pretraining a score network as specified in the paper, and its score-based update remains computationally more demanding than a discriminator-based update like in GANs – since the score function takes values in the data space, while the discriminator output is scalar. Training Discriminator Flow requires sampling at every step from the generating differential equation of Equation (20). This makes its training iterations slower than both diffusion models (which do not require resampling through the differential equation) and GANs (which have fast sampling).

Besides the cost of individual training iterations, the total temporal cost of training also depends on the number of iterations, which we specify in Appendix D.3.

# B Algorithmic Details

We detail in this section some aspects of the Score GAN and Discriminator Flow algorithms that were described at a high level in Section 4.

## B.1 Score GANs

Algorithm 1 involves two major steps: line 3 performs denoising score matching for the generated distribution, and line 5 updates the generator parameters using the resulting gradient flow in Equation (19). We describe some implementation tricks for these steps in the following subsections.

### B.1.1 Denoising Score Matching

Denoising score matching was described in line 3 of Algorithm 1, as originally used by Song & Ermon (2019). Following best practices later introduced by Karras et al. (2022), instead of directly training a network $s_\phi^\rho$ to estimate the score, we instantiate and train a denoising network $d_\phi^\rho$ using the following update:

$$\phi \leftarrow \phi - \lambda \nabla_\phi \left\| d_\phi^\rho(x^\sigma, \sigma) - x \right\|_2^2, \tag{23}$$

where $d_\phi^\rho$ is implemented as a U-Net (Ronneberger et al., 2015), with additional input-output skip connections for better preconditioning (Karras et al., 2022). We then use $d_\phi^\rho$ to compute the estimated score:

$$s_\phi^\rho(x, \sigma) = \frac{d_\phi^\rho(x, \sigma) - x}{\sigma^2}. \tag{24}$$

We use the same tricks to pretrain the score model of the data distribution, $s_\psi^{p_{\text{data}}}$.

### B.1.2 Generator Training

Line 5 of Algorithm 1 indicates how to update the generator parameters, following Equation (11). In order to facilitate its implementation in deep learning frameworks, we instead use the equivalent

**Algorithm 3:** Training iteration for Discriminator Flow (detailed).

---

**Input:** Batch size $B \in \mathbb{N}^*$, number of steps $N \in \mathbb{N}^*$, initial distribution $\pi = \rho_0$, gradient strength $\eta \in \mathbb{R}_+$, previous discriminator $f_\phi \colon \mathbb{R}^D \times \mathbb{R} \to \mathbb{R}$.

**Output:** Updated discriminator $f_\phi$.

1  **for** $b = 1$ **to** $B$ **do**                                       `// In parallel`

2     $x^b \sim p_{\mathrm{data}}, x_0^b \sim \pi;$                                 `// Initialization`

      `// Partial generation`

3     **for** $i = 0$ **to** $N - 1$ **do**                           `// Solve Equation (20)`

4         $x_{i+1/N}^b \leftarrow x_{i/N}^b - \frac{\eta}{N} \nabla_{x_{i/N}^b} \left[ (c \circ f_\phi)\left(x_{i/N}^b, \frac{i}{N}\right) \right];$

5     $i_b \sim \mathcal{U}(\llbracket 0, N-1 \rrbracket);$                          `// Select random step`

  `// Train discriminator `$f_\phi$` at the chosen random steps, cf. Equation (15)`

6  $\phi \leftarrow \mathrm{GA}_\phi \left\{ \mathcal{L}_\mathrm{d}\left( f_\phi\left(\cdot, \frac{i_b}{N}\right); \mathcal{U}\left(\left\{x_{i_b/N}^b\right\}_{b \in \llbracket 1, B \rrbracket}\right), \mathcal{U}\left(\left\{x^b\right\}_{b \in \llbracket 1, B \rrbracket}\right) \right) \right\}$

---

weight update:

$$\theta \leftarrow \theta + \eta \nabla_\theta \left[ g_\theta(z)^\top \mathrm{StopGradient}\left( s_\psi^{p_{\mathrm{data}}}\big(g_\theta(z) + \sigma \varepsilon, \sigma\big) - s_\phi^\rho\big(g_\theta(z) + \sigma \varepsilon, \sigma\big) \right) \right], \quad (25)$$

that is, we optimize the loss $-g_\theta(z)^\top \mathrm{StopGradient}\left( s_\psi^{p_{\mathrm{data}}}\big(g_\theta(z) + \sigma \varepsilon, \sigma\big) - s_\phi^\rho\big(g_\theta(z) + \sigma \varepsilon, \sigma\big) \right)$.
We also adapt the weight update as follows. While keeping the same $h_\rho$ as in Equation (19), we multiply it in the weight update by $\sigma$ (which amounts to changing the noise level sampling distribution $p_\sigma$ accordingly). When implemented with a denoiser network as described in Appendix B.1.1, the score estimation $s_\phi^\rho(x, \sigma)$ can be numerically unstable for small $\sigma$s when using Equation (24). This explains why Karras et al. (2022) consistently multiply scores by $\sigma$, which is a choice that we follow for Score GANs. Furthermore, we can generalize Score GANs to other functionals $h_\rho$ encompassing the EDM formulation of Equation (9), right:

$$h_{\rho_t} = \mu_{p_{\mathrm{data}}} \log[p_{\mathrm{data}} \star k_{\mathrm{RBF}}^\sigma] - \mu_\rho \log[\rho_t \star k_{\mathrm{RBF}}^\sigma], \quad (26)$$

where $\mu_{p_{\mathrm{data}}}$ and $\mu_\rho$ are constant hyperparameters (e.g., for EDM, $\mu_{p_{\mathrm{data}}} = 2$ and $\mu_\rho = 1$).

Overall, in practice we implement the generator weight update of line 5 from Algorithm 1 as:

$$\theta \leftarrow \theta + \eta \nabla_\theta \left[ \sigma \cdot g_\theta(z)^\top \mathrm{StopGradient}\left( \mu_{p_{\mathrm{data}}} s_\psi^{p_{\mathrm{data}}}\big(g_\theta(z) + \sigma \varepsilon, \sigma\big) - \mu_\rho s_\phi^\rho\big(g_\theta(z) + \sigma \varepsilon, \sigma\big) \right) \right]. \tag{27}$$

## B.2 Discriminator Flows

In Algorithm 3 we provide a detailed implementation of the high-level Algorithm 2 for Discriminator Flows, including how batching and differential equation discretization are handled. For the latter, we use the Euler method with a uniform temporal grid, with $N$ steps, in $[0, 1]$. While higher-order solvers and more optimal temporal grid choice could have been used, as in EDM (Karras et al., 2022), we avoid using any refinement of time discretization for Discriminator Flows for the sake of simplicity. For batch-parallel execution on GPUs, we solve the differential equation of Equation (20) over $[0, 1]$ for all batch samples, and then select a random time for each sample to compute the discriminator loss and update its parameters.

## C Further Experiments

In this section we present additional experiments that will help the reader develop better intuition on the behavior of both Score GANs and Discriminator Flows in practice.

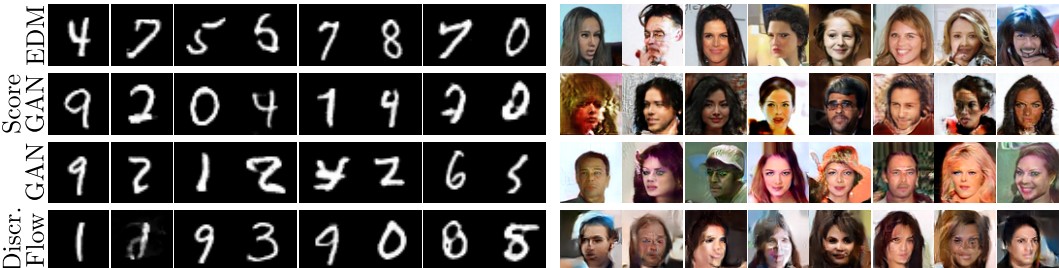

Figure 3: Uncurated samples from studied models trained on CelebA and MNIST.

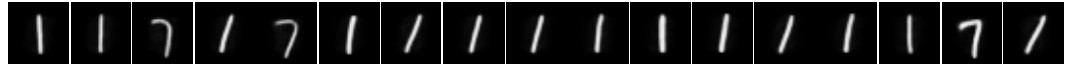

Figure 4: Uncurated samples of a Score GAN variant, which shows strong mode collapse on MNIST. Note that the Score GAN parameters were intentionally chosen to obtain this behavior.

## C.1    Additional Samples

Figure 3 shows additional samples for all four tested models. Furthermore, for better visualization purposes, our public repository https://github.com/White-Link/gpm includes animated images illustrating the generation process for EDM and Discriminator Flows, as illustrated in Figure 1.

## C.2    Mode Collapse on Score GANs

A widely known issue for GANs, identified soon after their introduction, is known as mode collapse (Goodfellow, 2016), which occurs when the generator only covers a fraction of the generated distribution. Interestingly, we observe the same phenomenon in Score GANs. We illustrate this with an extreme example in Figure 4, where we intentionally change parameters in our original model in order to induce the mode collapse issue. We induce mode collapse by choosing $\mu_{p_{\text{data}}} = 2$ and $\mu_\rho = 1$ in Equation (27) (i.e., EDM parameters instead of NCSN parameters in the original model).

Since mode collapse is absent from the generator-less particle models that we tested (the score models on which Score GANs are based, and also Discriminator Flows), this observation suggests that mode collapse is primarily caused by the generator. This is in line with previous theoretical and empirical findings identifying the generator as the cause of mode collapse (Tanielian et al., 2020; Durr et al., 2022).

## C.3    Latent Interpolations

We provide examples of interpolations on MNIST for all four tested models in Figure 5. We tested four interpolation methods on Gaussian priors considered by Leśniak et al. (2018): linear, spherical, Cauchy-linear, and spherical Cauchy-linear. We reached the same conclusion for each of these methods and thus only show the result of the most visually appealing one, spherical Cauchy-linear.

We notice that the generator-based models, Score GANs and GANs, show smoother transitions between generated images than the particle models EDM and Discriminator Flows, for which abrupt changes of digit identity and shape can be seen between consecutive interpolation steps. This confirms that interacting particle models (generator-based) can perform feature learning via their smaller latent space, allowing for smoother generation, while particle models operating in the data space are less prone to such phenomenon. In this regard, Score GANs and Discriminator Flows are no different than their parent method in the same model category.

## C.4    Alternating Updates of Generator and Score in Score GANs

There are $K$ steps of score updates per generator update in Score GANs, similarly to discriminators in GANs. Accordingly, $K$ is an important parameter in Score GANs. First of all, like in GANs, the tuning of $K$ heavily depends on the ratio $r = \frac{\lambda}{\eta}$ between the learning rates of the score network and

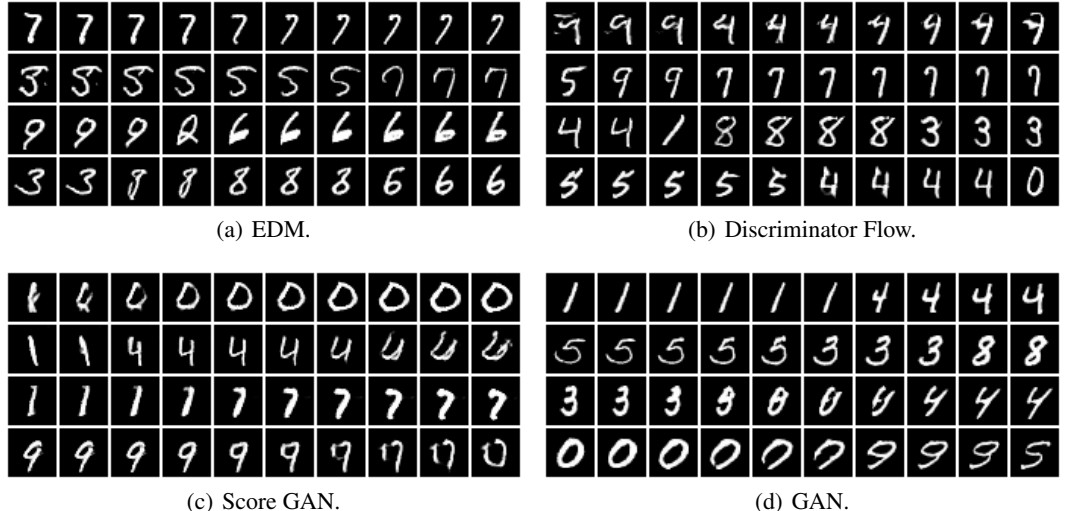

(a) EDM.

(b) Discriminator Flow.

(c) Score GAN.

(d) GAN.

Figure 5: Interpolations (row-wise) in the latent space of the studied models (uncurated samples) at regular intervals.

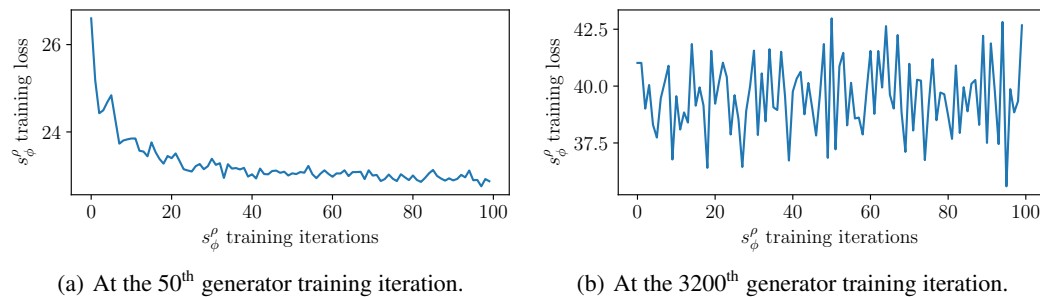

(a) At the 50th generator training iteration.

(b) At the 3200th generator training iteration.

Figure 6: Evolution on MNIST of the training loss in-between two generator updates of the score of the generated distribution $s_\phi^\rho$, for $K = 100$ and equal score / generator learning rates $\lambda = \eta$.

the generator (Jelassi et al., 2022) – cf. Algorithm 1. A higher ratio may allow us to decrease the necessary number of steps $K$.

In our experiments on image data, we use $r \geq 2$ (Table 7). However, to gain more intuition empirically, we performed a set of experiments on MNIST with $\lambda$ such that $r = 1$, by making the number of steps $K$ vary from 1 to 10. We obtained similar results across this range of values for $K$, close to the values reported in Table 3. This indicates that even low values of $K$ can provide a sufficient approximation of the score of the generated distribution.

To observe this qualitatively, we plot in Figure 6 the evolution of the score training loss in between generator updates for $K = 100$. At the beginning of training, $s_\phi^\rho$ needs around 20 updates to converge. Yet, after a small number of generator updates, it is already close to the optimum before its first update.

This confirms that the continuous update of the score of the generated distribution $s_\phi^\rho$, like a discriminator, coupled with a learning rate ratio $r > 1$, makes a small number of score updates $K$ between generator updates sufficient for adequate performance.

## C.5 Time Efficiency of Discriminator Flows

The qualitative experimental results in Section 4 suggest that Discriminator Flows, by learning a path towards the data distribution with a discriminator, may converge faster towards $p_{\text{data}}$ than diffusion

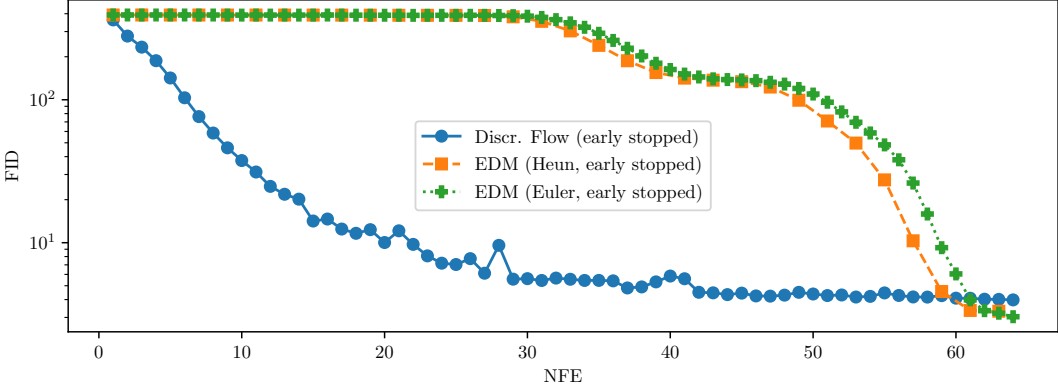

(a) Stopping the generative process of Equation (20) at an earlier time $T'$ than in training ($0 < T' < T = 1$).

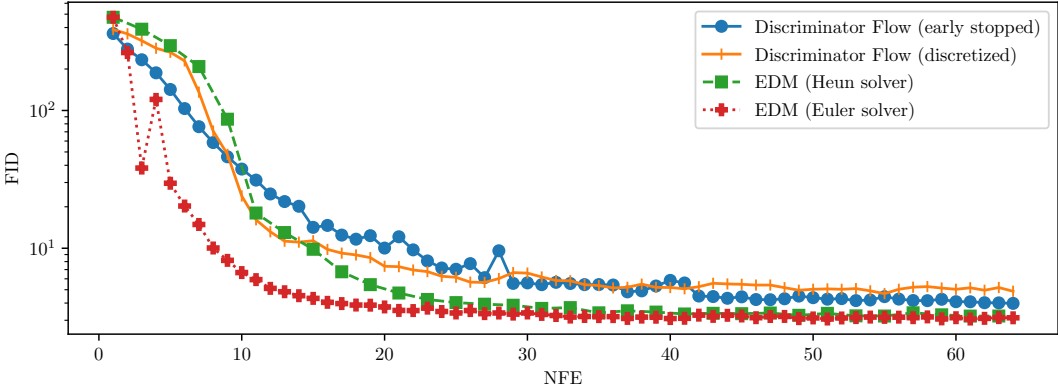

(b) Time efficiency comparison between alternative samplers of Discriminator Flows and EDM.

Figure 7: FID performance versus NFE (neural of function evaluations, i.e., the number of times $\nabla h_\rho$ is queried to produce a single image) for EDM and Discriminator Flow on MNIST. We test different NFE modulation methods for the Discriminator Flow, in order to evaluate the time efficiency of each method. We consider two standard differential equation solvers for EDM: Euler and Heun.

models, for which the path towards the data distribution is determined by the diffusion equation, which must be followed until the final time $T$.

We quantitatively confirm this observation in Figure 7(a), which displays the performance of EDM and Discriminator Flow on MNIST for intermediate generated distributions $\rho_t$, $t < T$, i.e., early stopping of the generation process. As expected given that intermediate generations from diffusion models are noisy, Discriminator Flows converge faster towards the data distribution. This raises the question of whether this faster convergence can yield a better efficiency for Discriminator Flows than for diffusion models. We investigate this claim in the following.

Time efficiency in diffusion models is usually measured, as done by Karras et al. (2022), in terms of generative performance versus number of neural function evaluations (NFEs): the most costly operation is the repeated evaluation of the score network throughout the diffusion process. We then study the efficiency of EDM and Discriminator Flow using this metric.

Since Discriminator Flows are based on a discretized differential equation, we can measure the time efficiency of this model by decreasing the number of neural function evaluations (NFEs) in two ways: by using a larger time discretization of Equation (20) or by an early stop of the generative process at time $T' < T$ as already described above. For the first alternative, we train the evaluated model with 128 generative steps and $\eta = 1$; for the second one, we train the evaluated model with 64 generative steps and $\eta = 2$, i.e., with half as many generative steps as the first one but with a doubled velocity, c.f. Algorithm 3. We compare both alternatives against two standard inference methods for EDM

based on the (first-order) Euler solver (Kloeden & Platen, 1992), and the (second-order) Heun / EDM sampler (Karras et al., 2022). We show the results obtained for MNIST in Figure 7(b).

Overall, the Discriminator Flow model is comparable to the second-order version EDM; however, both remain outperformed by the first-order version of EDM. Surprisingly, the first-order version of EDM is actually more efficient in terms of NFE than the second-order version for low NFE values. This baseline was not considered by Karras et al. (2022), so this is a new observation; we confirmed this phenomenon using the official implementation of EDM on CIFAR10 (Krizhevsky, 2009). This behavior is especially visible in our case as we test it on a simpler dataset.

Therefore, additional experiments are necessary to form a conclusion on the relative efficiency of Discriminator Flows w.r.t. diffusion models. We stress that the current results are achieved with Discriminator Flows discretizing Equation (20) on a regular temporal grid, while EDM discretize time for both first- and second-order methods using a custom temporal grid that improves discretization performance. Finally, we believe that further tuning of Discriminator Flows could significantly improve its efficiency: while the final image appears quickly after a few steps as illustrated in Figure 1, some imperceptible residual noise still needs to be eliminated in the remaining steps. This noise prevents steady convergence towards $p_{\text{data}}$; removing it one of the main directions of improvement for Discriminator Flows.

### C.6 Experiments on Gaussians

As a toy example, we train all considered models (Discriminator Flows, GANs, EDM, and Score GANs) on synthetic samples generated from a two-dimensional mixture of Gaussian distributions. We provide a visualization of the particle evolution w.r.t. training and inference time in our repository `https://github.com/White-Link/gpm`, as well as in Figures 8 to 12. See the figure captions for more information.

We again observe that Discriminator Flows converge faster than EDM towards the data distribution.

## D  Experimental Details

We provide all details in this section that are necessary to reproduce our experiments. Our Python source code (tested on version 3.10.4), based on PyTorch (Paszke et al., 2019) (tested on version 1.13.1), is available as open source at `https://github.com/White-Link/gpm`.

### D.1  Datasets and Evaluation Metric

**MNIST.**   MNIST is a standard dataset introduced in LeCun et al. (1998), with no clear license to the best of our knowledge, composed of monochrome images of hand-written digits. Each MNIST image is single-channel, of size $28 \times 28$. We preprocess MNIST images by extending them to $32 \times 32$ frames (padding each image with black pixels), in order to better fit as inputs and outputs of standard convolutional networks. We linearly scale pixels values so that they lie in $[-1, 1]$. MNIST is comprised of a training and testing dataset, but no validation set; we create one for each model training by randomly selecting $10\%$ of the training images.

**CelebA.**   CelebA (Liu et al., 2015) is a dataset composed of celebrity pictures. Its license permits use for non-commercial research purposes. Each CelebA image has three color channels, and is of size $178 \times 218$. We preprocess these images by center-cropping each to a square image and resizing to $64 \times 64$ with a Lanczos filter. We linearly scale pixels values so that they lie in $[-1, 1]$.

**Gaussians.**   Our Gaussian dataset is composed of a mixture of 5 two-dimensional Gaussian distributions with standard deviation $\frac{1}{2}$, with means evenly spaced over a circle of radius 5. Training, validation, and testing datasets all consist of i.i.d. samples from this mixture.

**FID.**   Throughout the paper we use the Fréchet Inception Distance (FID, Heusel et al., 2017) to measure the generative performance of the models we consider. In our code we use the PyTorch implementation of TorchMetrics (Skafte Detlefsen et al., 2022).

### D.2 Hyperparameters

We summarize the model hyperparameters used during training in Tables 4 to 7. See our code for more information. We further discuss some aspects of our implementation choices in the remainder of this subsection.

**Networks.** Beyond multi-layer perceptrons (MLP) and EDM's U-Nets, we use three kinds of model architectures: DCGAN (Radford et al., 2016), a ResNet (Kang et al., 2022) based on SAGAN with self-attention (Zhang et al., 2019), and FastGAN (Liu et al., 2021). We adapt these models in the following ways:

- For MNIST images, we changed the first (respectively, last) convolution of the DCGAN discriminator (respectively, generator) to adapt it for $32 \times 32$ inputs (respectively, outputs).

- We adapted FastGAN to operate on images of size $32 \times 32$ and $64 \times 64$ by removing layers with higher resolutions.

- We added a bias to the first convolutional layers of discriminators, which did not have one.

- We enhanced these models so that they can accept a vectorial embedding of time $t$ for Discriminator flows, by modulating most of their convolution outputs (before the activation) channel-wise with an affine transformation. The affine transformation parameters are the outputs of a MLP of depth 2, with SiLU activations (Hendrycks & Gimpel, 2016) and a hidden width that is twice as large as the input time embeddings.

**Final generator activation.** Since image pixel values lie in $[-1, 1]$, we add a final hyperbolic tangent activation to the output of the generators operating on image data.

**Score GAN noise distribution $p_\sigma$.** Since during training Score GANs mimic the inference procedure of a diffusion model, we choose to sample $\sigma$ during Score GAN training time following the schedule chosen by EDM at inference time (Karras et al., 2022). In practice, we choose a minimal value $\sigma_{\min}$, a maximal value $\sigma_{\max}$, and an interpolation parameter $\rho$. To sample from $p_\sigma$, we first sample an interpolation value $\alpha \sim \mathcal{U}\big([0, 1]\big)$, and then compute $\sigma$ as:

$$\sigma = \left( \sigma_{\max}^{\frac{1}{\rho}} + \alpha \left( \sigma_{\min}^{\frac{1}{\rho}} - \sigma_{\max}^{\frac{1}{\rho}} \right) \right)^{\rho}. \tag{28}$$

Note that here $\rho$ denotes a scalar hyperparameter following the EDM notation, and not the generated distribution as in the main paper.

**Usual generative modeling tricks.** For simplicity in our proof-of-concept experiments, we avoided using standard performance improvement tricks such as exponential moving average and truncation tricks (Brock et al., 2019), EDM sampling tricks (Karras et al., 2022), or spectral normalization (Miyato et al., 2018).

### D.3 Compute

For all experiments we use one or two Nvidia V100 GPUs with CUDA 11.8. When using two V100 Nvidia GPUs, the training time of both Discriminator Flow and Score GAN models is at most one day for the largest dataset (CelebA).

As a means to evaluate each model's efficiency, we provide the approximate amount of time we used to train the tested models over MNIST on one NVIDIA V100 GPU:

- for Discriminator Flow, 24 hours;

- for Score GAN, 10 hours (excluding the pretrained diffusion model);

- for EDM, 6 hours;

- for GAN, 1 hour.

Table 4: Chosen hyperparameters for Discriminator Flows for each dataset. GP stands for Gradient Penalty (Gulrajani et al., 2017). IPM stands for Integral Probability Metric (Müller, 1997). BN stands for Batch Normalization (Ioffe & Szegedy, 2015).

| Hyperparameters | Gaussians | MNIST | CelebA |
|---|---|---|---|
| GAN loss | Non-saturating vanilla | IPM | IPM |
| $a$ (Equation (15)) | $\log(1 - \mathrm{sigmoid})$ | id | id |
| $b$ (Equation (15)) | $-\log \mathrm{sigmoid}$ | id | id |
| $c$ (Equation (15)) | $-\log \mathrm{sigmoid}$ | id | id |
| $\mathcal{R}$ (Equation (15)) | 0 | GP | GP |
| GP strength | 0 | 0.04 | 0.05 |
| GP center | — | 0 | 0 |
| $\pi$ (Equation (1)) | $\mathcal{N}(0, I_D)$ | | |
| $\eta$ (Algorithm 3) | 20 | $[2, 1]$ | 2 |
| $N$ (Algorithm 3) | 56 | $[64, 128]$ | 25 |
| Network architecture | MLP | DCGAN | ResNet |
| Width | 512 | 64 | 64 |
| Activation | Leaky ReLU, negative slope of $-0.2$ | | |
| Depth | 4 | — | — |
| BN | No | | |
| Initialization | Normal | Orthogonal | Normal |
| Initialization gain | 0.02 | 1.41 | 0.02 |
| Time embedding type | Fourier | | |
| Frequency scale | 16 | | |
| Time embedding size | 128 | | |
| Batch size | 128 | 128 | 64 |
| Number of optimization steps | 4000 | 200 000 | 200 000 |
| Optimizer | Adam (Kingma & Ba, 2015) | | |
| Learning rate | 0.0002 | | |
| $(\beta_1, \beta_2)$ | $(0.5, 0.999)$ | | |
| Model selection | — | Best validation FID | |
| Validation frequency | — | 1000 | |
| Number of validation samples | — | 6000 | 1000 |

# E   Broader Impacts

Our work aims to better understand recent generative modeling methods. As such, from a practical point of view, our work shares much of the impact of other work in this domain. While the models we propose do not yet have the quality required for broad application, generative models have a wide range of potential positive and negative impacts. Some of the positive impacts include enabling faster and more accurate natural language processing (Li et al., 2022), improving automated tasks like summarization (Liu et al., 2018), and automating certain aspects of content creation (Nichol et al., 2022). However, generative models are also susceptible to producing undesirable output, such as unethical text, adversarial attacks (Wang et al., 2021), and malicious manipulation of data. These models also have the potential to exacerbate issues such as bias and discrimination in AI systems (Lucy & Bamman, 2021), enabling the creation of more effective fake text and videos, and expanding the scope and complexity of cyberattacks that can be launched by malicious actors (Seymour & Tully, 2018). For a thorough discussion on the potential dangers of deepfakes, see Fallis (2021).

Discussions and debates around the responsible use of generative models and the potential broader impacts of deploying them more widely are still ongoing. We hope that our principled framework aimed at improving our understanding of generative models will contribute to these discussions and to better control of such models.

Table 5: Chosen hyperparameters for EDM for each dataset. Cf. Karras et al. (2022) and our code for more details.

| Hyperparameters | Gaussians | MNIST | CelebA |
|---|---|---|---|
| $\sigma_{\min}$ | | 0.002 | |
| $\sigma_{\max}$ | | 40 | |
| $\sigma_{\text{data}}$ | | 0.5 | |
| $\rho$ | | 7 | |
| Equation & Solver | Heun solver on the deterministic ODE of Equation (8) | | |
| Number of solver steps | 7 | 32 | 25 |
| Network architecture | MLP | EDM | EDM |
| Width | 512 | 16 | 128 |
| Number of residual blocks | — | 1 | 2 |
| Dropout | — | 0.13 | 0.1 |
| Depth | 4 | — | — |
| Activation | Leaky ReLU, slope of $-0.2$ | — | — |
| Initialization | Uniform Kaiming for convolutions, unit weight and zero bias for group normalization layers, otherwise PyTorch default | | |
| Time embedding type | Fourier | Positional | Positional |
| Frequency scale | 16 | — | — |
| Time embedding size | 128 | 256 | 256 |
| Batch size | 128 | 128 | 64 |
| Number of optimization steps | 10 000 | 500 000 | 100 000 |
| Optimizer | Adam (Kingma & Ba, 2015) | | |
| Learning rate | 0.0002 | | |
| $(\beta_1, \beta_2)$ | $(0.9, 0.999)$ | | |
| Model selection | — | | |

Table 6: Chosen hyperparameters for GANs for each dataset. Hinge refers to Hinge GANs (Lim & Ye, 2017).

| Hyperparameters | Gaussians | MNIST | CelebA |
|---|---|---|---|
| GAN loss | Non-saturating vanilla | | Hinge |
| Number of discriminator steps per generator update | | 1 | |
| Latent space size | | 128 | |
| $p_z$ (Definition 2) | | $\mathcal{N}(0, I_d)$ | |
| Discriminator architecture | MLP | DCGAN | FastGAN |
| Width | 512 | 64 | 32 |
| Activation | Leaky ReLU, negative slope of $-0.2$ | | |
| Depth | 4 | — | — |
| BN | No | Yes | Yes |
| Generator architecture | MLP | DCGAN | FastGAN |
| Width | 512 | 64 | 32 |
| Activation | Leaky ReLU, slope of $-0.2$ | ReLU | Leaky ReLU, slope of $-0.2$ |
| Depth | 4 | — | — |
| BN | No | Yes | Yes |
| Initialization | | Normal | |
| Initialization gain | | 0.02 | |
| Batch size | 128 | 128 | 64 |
| Number of optimization steps | 1900 | 10 000 | 100 000 |
| Optimizers | Adam (Kingma & Ba, 2015) | | |
| Learning rate | | 0.0002 | |
| $(\beta_1, \beta_2)$ | | $(0.5, 0.999)$ | |
| Model selection | — | Best validation FID | |
| Validation frequency | — | 1000 | |
| Number of validation samples | — | 6000 | 1280 |

Table 7: Chosen hyperparameters for Score GANs for each dataset.

| Hyperparameters | Gaussians | MNIST | CelebA |
|---|---|---|---|
| $\sigma_{\min}$ (Appendix D.2) | 0.1 | 0.32 | 0.32 |
| $\sigma_{\max}$ (Appendix D.2) | 10 | 40 | 40 |
| $\rho$ (Appendix D.2) | | 3 | |
| $K$ (Algorithm 1) | 10 | 1 | 4 |
| Latent space size | | 128 | |
| $p_z$ (Definition 2) | | $\mathcal{N}(0, I_d)$ | |
| Data score $s_\psi^{p_{\text{data}}}$ | | EDM from Table 5 | |
| Gen. score $s_\phi^\rho$ architecture | MLP | EDM | EDM |
| $\sigma_{\text{data}}$ | | 0.5 | |
| Width | 512 | 64 | 128 |
| Number of residual blocks | — | 2 | 2 |
| Dropout | — | 0.13 | 0 |
| Depth | 4 | — | — |
| Activation | Leaky ReLU, slope of $-0.2$ | — | — |
| Initialization | | EDM from Table 5 | |
| Generator architecture | MLP | DCGAN | FastGAN |
| Width | 512 | 64 | 32 |
| Activation | | ReLU | |
| Depth | 4 | — | — |
| BN | No | Yes | Yes |
| Initialization | PyTorch default | Normal | Orthogonal |
| Initialization gain | — | 0.02 | 1.41 |
| Batch size | 128 | 256 | 32 |
| Number of generator optimization steps | 3150 | 100 000 | 150 000 |
| Optimizers | | Adam (Kingma & Ba, 2015) | |
| Score learning rate | 0.0002 | 0.001 | 0.0004 |
| Generator learning rate | | 0.0002 | |
| $(\beta_1, \beta_2)$ | | $(0.9, 0.999)$ | |
| Model selection | — | Best validation FID | — |
| Validation frequency | — | 2500 | — |
| Number of validation samples | — | 6000 | — |

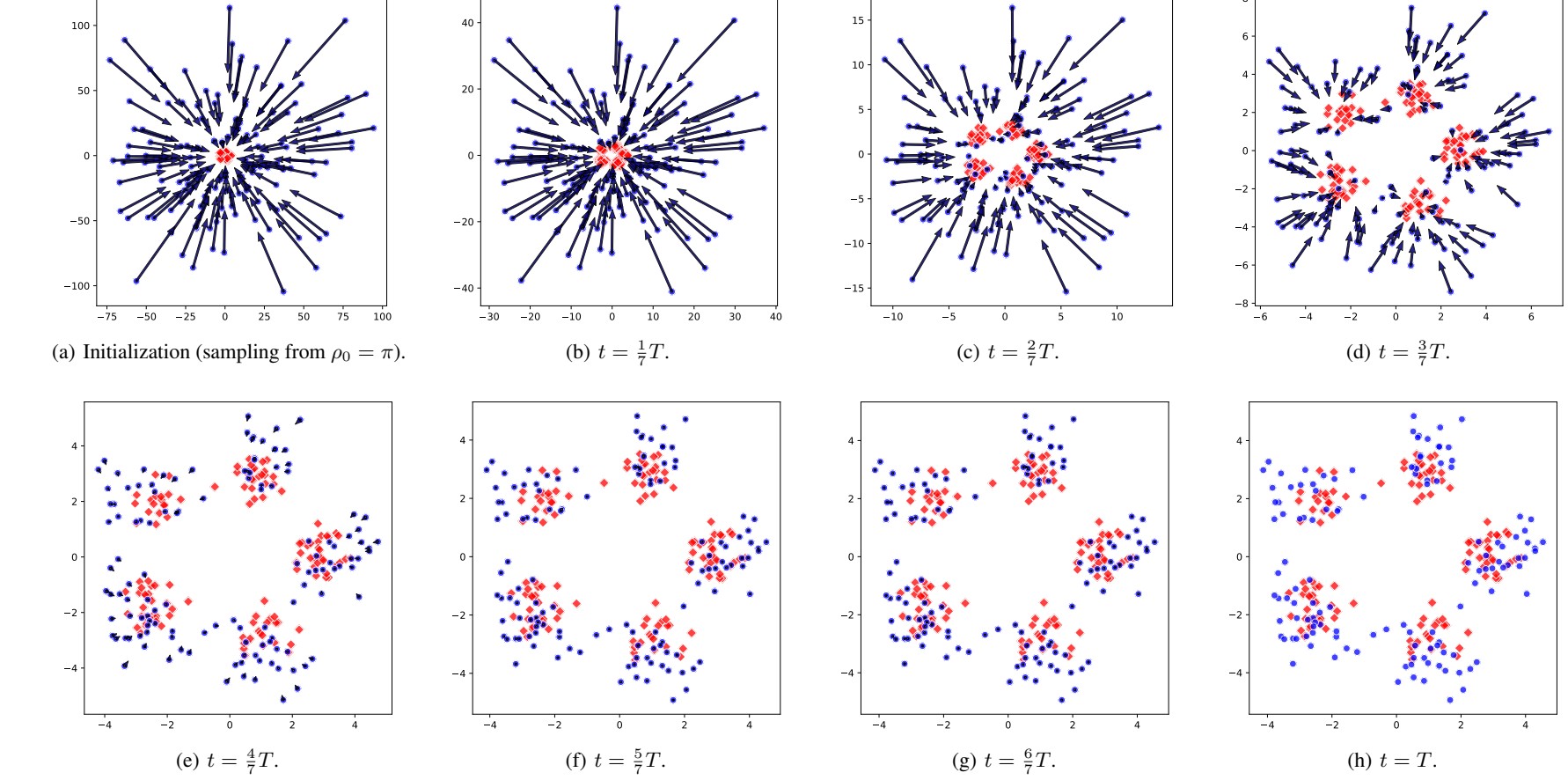

(a) Initialization (sampling from $\rho_0 = \pi$).     (b) $t = \frac{1}{7}T$.     (c) $t = \frac{2}{7}T$.     (d) $t = \frac{3}{7}T$.

(e) $t = \frac{4}{7}T$.     (f) $t = \frac{5}{7}T$.     (g) $t = \frac{6}{7}T$.     (h) $t = T$.

Figure 8: Sampling steps for EDM on a Gaussian mixture; 128 samples are shown for both the generated (●) and the data (◆) distributions. The second-order solver was used with 13 NFEs. Arrows show the gradients $\nabla h_{\rho_t}$ associated with each generated sample, corresponding to the direction provided by the score function, cf. Equation (8).

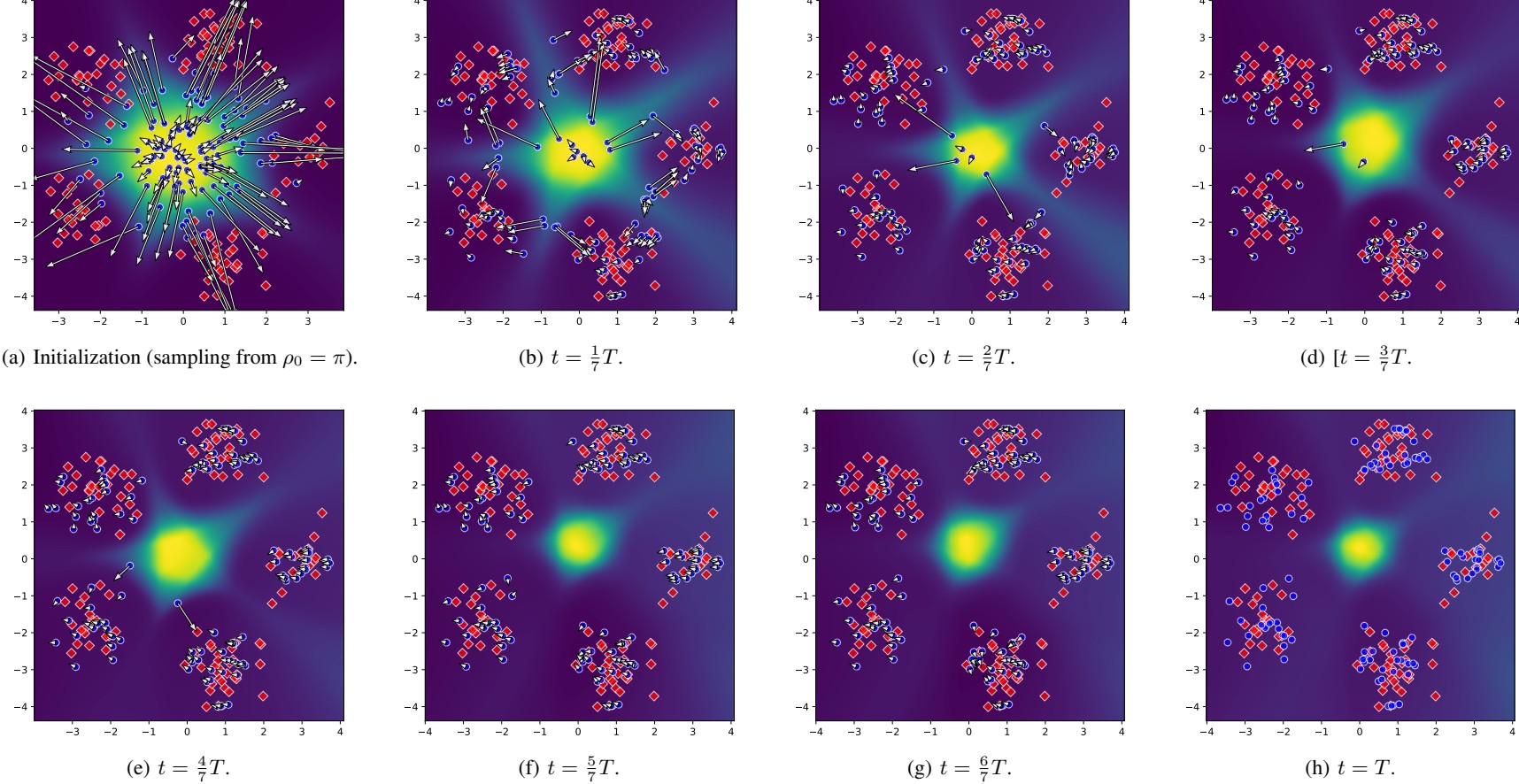

(a) Initialization (sampling from $\rho_0 = \pi$).

(b) $t = \frac{1}{7}T$.

(c) $t = \frac{2}{7}T$.

(d) $[t = \frac{3}{7}T$.

(e) $t = \frac{4}{7}T$.

(f) $t = \frac{5}{7}T$.

(g) $t = \frac{6}{7}T$.

(h) $t = T$.

Figure 9: Sampling steps for Discriminator Flows on a Gaussian mixture; cf. Figure 8. Our chosen discretization yields 14 NFEs. The colored background represents the pointwise generator loss function $-h_{\rho_t} = -c \circ f_{\rho_t}$ (darker is lower, renormalized for every snapshot), from which the particle gradients, shown in the figures, are derived.

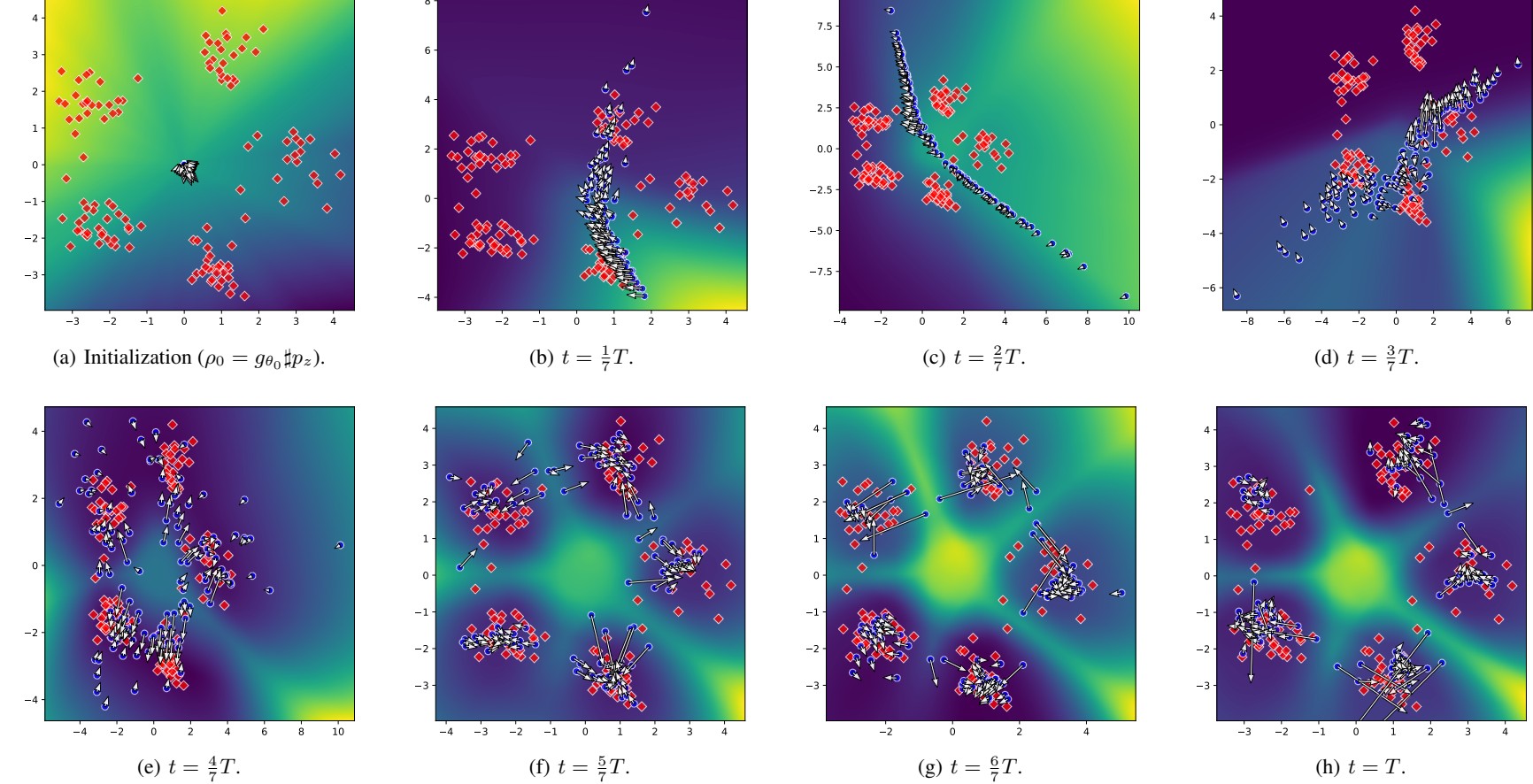

(a) Initialization ($\rho_0 = g_{\theta_0} \sharp p_z$).    (b) $t = \frac{1}{7}T$.    (c) $t = \frac{2}{7}T$.    (d) $t = \frac{3}{7}T$.

(e) $t = \frac{4}{7}T$.    (f) $t = \frac{5}{7}T$.    (g) $t = \frac{6}{7}T$.    (h) $t = T$.

Figure 10:    Training snapshots of a GAN on a Gaussian mixture; cf. Figure 8. Here time $t$ represents training time, and $T$ is the end of training. The colored backgrounds represent the pointwise generator loss function $-h_{\rho_t} = -c \circ f_{\rho_t}$ (darker is lower, renormalized for every snapshot), from which the particle gradients, shown in the figures, are derived and then fed to the generator following Equations (11) and (13).

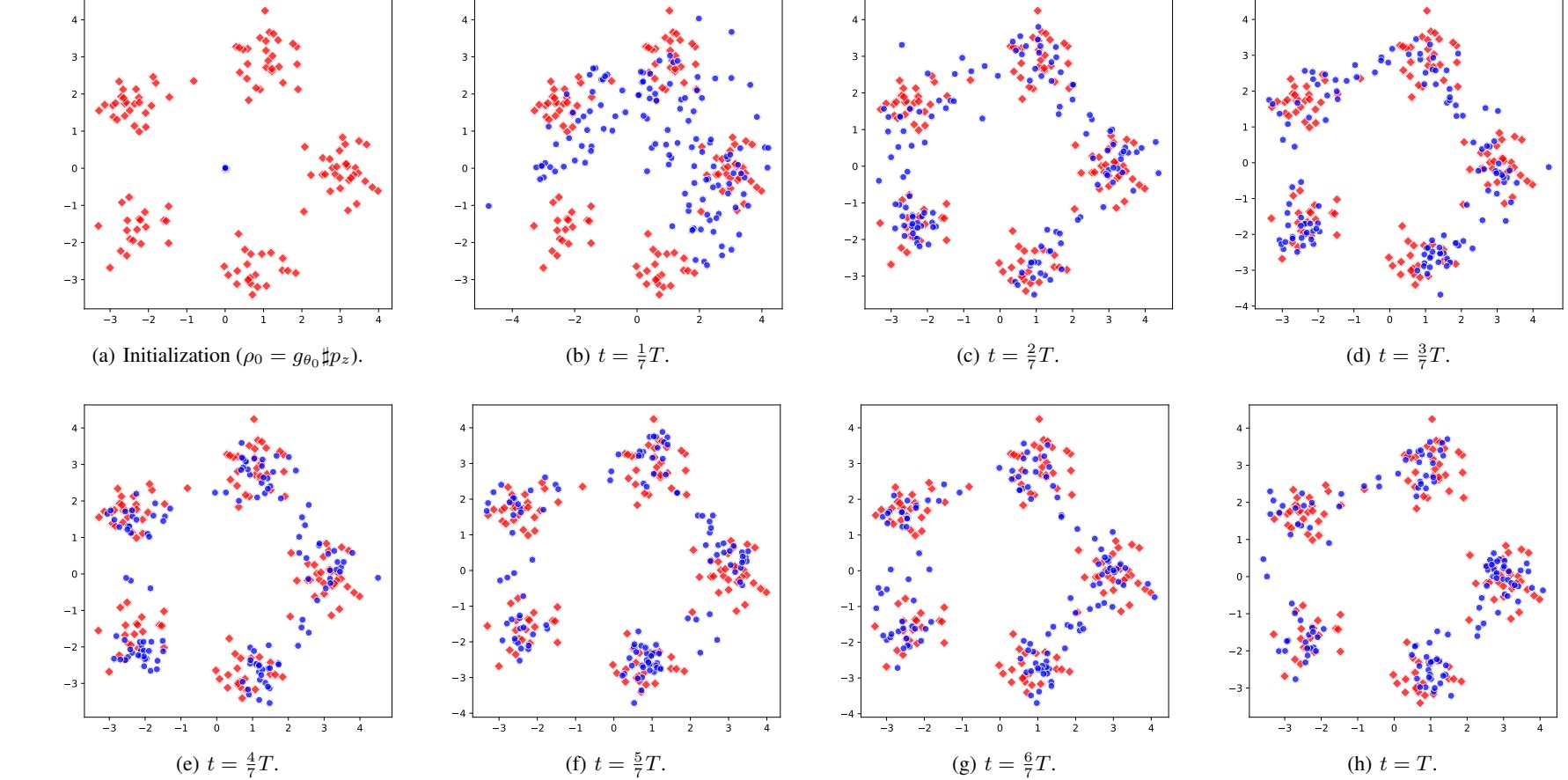

(a) Initialization ($\rho_0 = g_{\theta_0} \natural p_z$).     (b) $t = \frac{1}{7}T$.     (c) $t = \frac{2}{7}T$.     (d) $t = \frac{3}{7}T$.

(e) $t = \frac{4}{7}T$.     (f) $t = \frac{5}{7}T$.     (g) $t = \frac{6}{7}T$.     (h) $t = T$.

Figure 11: Training snapshots of a Score GAN on a Gaussian mixture; cf. Figure 8. Here time $t$ represents training time, and $T$ is the end of training.

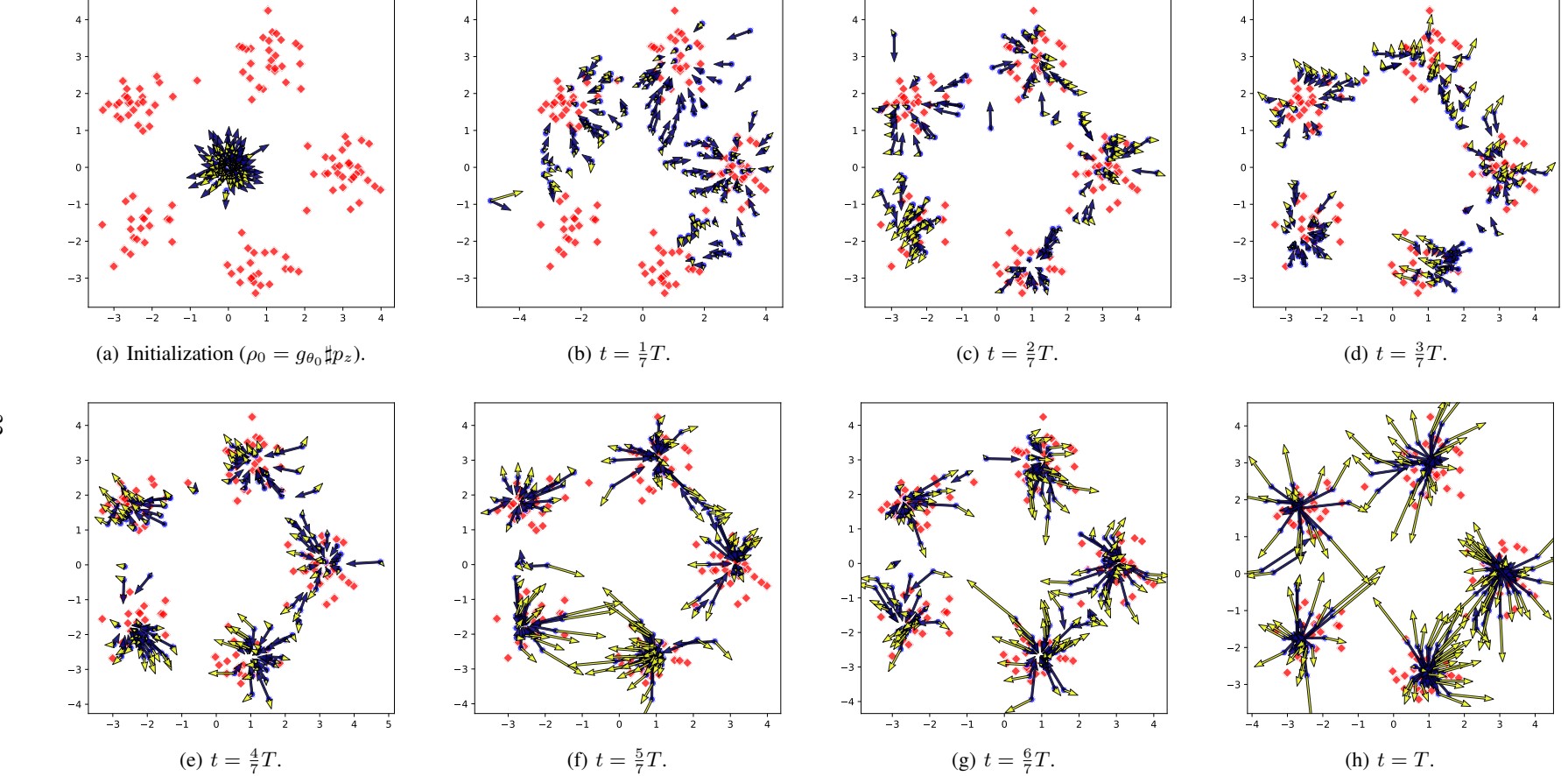

(a) Initialization ($\rho_0 = g_{\theta_0}\sharp p_z$). (b) $t = \frac{1}{7}T$. (c) $t = \frac{2}{7}T$. (d) $t = \frac{3}{7}T$.

(e) $t = \frac{4}{7}T$. (f) $t = \frac{5}{7}T$. (g) $t = \frac{6}{7}T$. (h) $t = T$.

Figure 12: Training snapshots of a Score GAN on a Gaussian mixture, identical to Figure 11, but with generated samples perturbed by Gaussian noise of standard deviation $\sigma = 0.2$. Arrows show the gradients $\nabla h_{\rho_t}$ received by the generated particles at this noise level, corresponding to Equation (19), split into the data score (in blue) and minus the score of the generated distribution (in yellow). They are then fed to the generator following Equations (11) and (13).