# OpenReview forum: "Unifying GANs and Score-Based Diffusion as Generative Particle Models"
_NeurIPS.cc/2023/Conference — NeurIPS 2023 poster_

### Official Review · Reviewer_6P1y · 2023-06-26

**Soundness:** 3 good
**Presentation:** 3 good
**Contribution:** 4 excellent
**Rating:** 5
**Confidence:** 5

**Summary:**

In this paper, the authors consider the task of generative modeling as one of particle flow, and provide a unifying theory for score- and diffusion based generator-free models, and generator-based GANs. They propose the particle-flow models that encompassed diffusion and score-based methods such as NCSN or EDMs and subsequently like the approach to incorporate GAN losses to result in flows based on the GAN discriminators. The authors subsequently propose two novel training algorithms. One based on training GAN generators with the score of the data, and its own push-forward distribution score, and another, based on discriminator-based flow. While the performance is not state-of-the-art, given the current interest in diffusion based models, a unifying theory that brings to light connections between GANs and flow based models is certainly interesting. I am convinced that these results could be used to leverage insights learned in the past decade of GAN training to potentially improve the performance of score-based models.

**Strengths:**

 - The proposed formulation is well presented and easy to understand. The overall concept on unifying score-based and GAN generator based modeling is clearly presented.
 - The contributions of the paper are generally made clear, presented as definitions and/or finding.
 - The limitations of the proposed method in achieving state-of-the-art performance are addresses head-on, which, in my opinion, is better, as it manages expectations on what the paper hopes to achieve — a clearer focus on theory, and not on showing SOTA.

**Weaknesses:**

- The training algorithm for Score GANs and discriminator flow could be better clarified. While the supplemental sections do address some of the questions I has, the algorithms / experimentation sections leave a good amount to be desired when it comes to understanding how training progresses (please see the __Questions__ section below for detailed concerns on experiments).
 - While most formulations are covered, I feel that a few additional comments could be made in the main text on how sampling/gridding schemes (of the SDE/PDE) serves to affect this formulation. For example, the authors could talk about the possible links between adjusted/unadjusted Langevin dynamics, and the NCSN/EDM formulations of PMs. does having $\sigma^{\prime}(t)\sigma(t)$ affect the discretization in any way. Across the framework, the authors stop at formulating the particle flow PDE, but maybe including comments on how discretizing this equation affects the understand could help. Could we drawn some connections with generator training (which, in some sense, is a “way to solve” to PM PDE) and how one discretized these SDE in diffusion/score-based models. Is this direction something the authors have considered?

 - The transition from equation (7) to (8) is treated a bit too casually, in my opinion. While (7) is an SDE, Karras et al., present a discussion on how the discretization could be split into a Probability flow ODE (PF-ODE), and a stochastic component. The formulation in this paper entirely ignores the stochastic component $\mathrm{d}W_t$, and places itself within the PF-ODE framework. Do the authors feel that this limit the scope of the proposed framework? For example, could they comment on what the implications of having a generator-based framework which solves a variant of (7), including  an exploratory noise term? Is it implicitly present in GAN training?

**Questions:**

Beyond the concerns remained in terms of the ODE formulation, I have a few additional concerns regarding the experimental formulation:

- I find it a bit hard to understand the overall picture in the training algorithms. I think the authors could add a bit more clarity on how both the score network for the generator in Score GANs, or the discriminator in discriminator flows are trained/used. First, in score GANs, if I understand correctly,  the score of the generator would be required at each training step $t$, to approximate the score of a sampling iteration index $t$, correct? The generator, as the authors also remark, is essentially distilling the entire sampling process of a score-based model, to a single generator sampling step. If so, it is not clear how Score GANs are trained. From Algorithm 1, I understand that $s_{\phi}^{\rho}$ is trained for K steps on a DMS loss, when the iterations on $t$ are replaced with noise levels $\sigma$, starting with $x=g_{\theta}(z)$. Is this $g_{\theta_t}(z)$? Does this mean that there are K steps of update per generator update. How large would K need to be to estimate the generator score accurately? Have the authors considered ablation experiments on this, at least say, on Gaussian data? I suspect the need for larger K as the data complexity increases.  Minor comment: Maybe mention what $p_{\sigma}$ is, in the main document as well (a one line summary of Lines L714-L720 might suffice).

 - I have similar concerns with the discriminator training.The presented explanation is not too clear. If my understand is correct, the explanations provided in the appendix indicate that the discriminator, provided with batches of data, considers random $t$-step Langevin sampling, and is trained such that the discriminator mimics this sampling behavior using a chosen loss? What step 6 in Algorithm 3 actually simplifies to, is unclear. It might help to add some intuition on what this loss amounts to minimizing, in the GAN context, (assuming the WGAN-GP loss, as the authors do, as per Table 4). Again, would this be an alternating scheme? Since $f_{\phi}$ is part of the sampler in lines 3-5, Algo. 3, would we need to reevaluate these samples after ever M steps of update on the discriminator? Is there an interplay between N and the number we would need for M? The authors explore this to a small degree in B.3, but I feel that the ablation experiments on these parameters could be a lot more exhaustive, or at least discussed in more detail.

- Did the authors also explore various embedding schemes for the time index in discriminator flows, and how different would the proposed discriminator be, from the formulation of Denoising Diffusion GANs (Xiao et al., ICLR 2022)? While the embedding is described in L707-711, it is not clear what Table 4 implies, regarding "Fourier" time embedding, or how frequency scale and embedding size matter. Lastly, while the time-efficiency analysis in B.3 does answer a few concerns I had on this regard, I’m not entirely convinced as to what the discriminator is acutely achieving, in comparison to a 2nd order Heun solver. I get that the performance of the discriminator flow is one par with second order methods, but can we say more about this? Would it be feasible to explore training the discriminator with samples generated via second order discretization?

 - __Minor__: (Figure 5, caption) … neural of function evaluations … — I’m not familiar with the original definition of the measure, but NFE should either be neural function evaluations or number of function evaluations. Equation (14),  on the right of the first equation, I do not see an occurrence of $z^{\prime}$. Should the summation be over $z^{\prime}$?


Overall, while I find that experimentation is lacking in some regards, such as ablation experiments, given that the paper is more theoretically aligned, it might at least help to provide insights or intuitive comments for variables/hyper-parameters where experiments are not included. I also think the authors would benefit from making the algorithms a bit clearer to understand, in terms of how the models are trained.

[1] Tackling the Generative Learning Trilemma with Denoising Diffusion GANs, Xiao et al., ICLR 2022

**Limitations:**

The paper is clear about its limitations on being an analytical exploration of GANs and diffusion-based approaches, and not an experiments one.

---

> ### Author Rebuttal · Authors · 2023-08-09
>
> We would like to thank the reviewer for their detailed feedback. Due to rebuttal size constraints, we answer the reviewer's main questions below. We will follow-up with the rest of the response during the discussion period.
>
> ## Training of Discriminator Flows
>
> Discriminator Flows do not attempt at mimicking Langevin sampling behavior. The principle of Discriminator Flows is to **make particles follow the same functional gradient $\nabla h_{\rho}$ as particles in GANs**, their corresponding interacting particle model, without the generator smoothing of Eq. (13). This corresponds to the gradient of the generator's loss that depends on the discriminator (cf. Eq. (20) which relates to Eq. (16) / Finding 5 in GANs).
>
> The most direct way to achieve this involves successively learning a discriminator per discretization step of Eq. (20), as follows.
> - Sample a batch of $(x_0^b)_b$ i.i.d. from the prior $\pi$.
> - For $i = 0$ to $N - 1$:
>   - $t = \frac{i}{N}$.
>   - Learn a new discriminator $f_{t}$ between the current generated distribution  $(x\_t^b)\_b$ and $p\_{\mathrm{data}}$ using a standard discriminator loss from Eq. (15) like WGAN-GP, i.e. like step 6 of Algorithm 3.
>   - Create the new batch of samples from the updated distribution following Eq. (20): $x_{t + \frac{1}{N}}^b = x_t^b - \frac{\eta}{N} \nabla_{x_t^b}(c \circ f_t)(x_t^b)$.
>
> However, this is impractical as the large number of independent networks to learn and the successive learning procedure make learning prohibitively slow.
>
> In our proposed version, we aim at simultaneously learning all discriminators to alleviate this issue. To this end, instead of learning multiple independent discriminators to cover all sampling times, we learn in Algorithm 3 a single time-parameterized network $f_{\phi}(\cdot, t)$ on all time steps at once. This may be seen as an alternating scheme, because updating of the discriminator naturally changes all intermediate generated distributions. As specified in step 2 of Algorithm 2, steps 3-4 of Algorithm 3 and lines 263-264, we do so after each discriminator update, i.e. $M = 1$.
>
> Overall, Discriminator Flows do not mimic a known sampling behavior a priori. The path that particles take depends on the chosen functions $a$, $b$, and $c$ parameterizing the losses of Eq. (15) and (16). Determining this path for general GAN losses such as WGAN-GP is an open problem. Nonetheless, it can be described under simplifying hypotheses. If $a$, $b$, and $c$ are chosen to implement an $f$-divergence GAN loss, then Discriminator Flows will implement a forward KL divergence gradient flow (Yi et al., 2023); if they correspond to an IPM GAN loss, then Discriminator Flows will implement a squared MMD gradient flow (Franceschi et al., 2022).
>
> This makes Discriminator Flows fundamentally different from Denoising Diffusion GANs (Xiao et al., 2022). On the one hand, Denoising Diffusion GANs train time-parameterized GANs to follow the inverse diffusion path. Hence, their discriminators are trained to discriminate between the intermediate generated distribution and the *noisy* data distribution. On the other hand, in Discriminator Flows, discriminators are trained to discriminate between the intermediate generated distribution and the *true*, non-noisy data distribution, thereby learning the path towards $p_{\mathrm{data}}$.
>
> ## Integration of the Stochastic Component in our Framework
>
> We thank the reviewer for this question, which allows us to develop this aspect of our framework. As the reviewer pointed out, for diffusion models, we focused on the probability flow of Eq. (9) -- which is the true equivalent of Eq. (7), instead of Eq. (8) as stated in most works on diffusion.
>
> We put aside the probabilistic formulation of Eq. (7) for purposes of clarity of exposition. Fortunately, it is possible to generalize our framework by taking into account the stochastic component of Eq. (7), as explained in the following. We will add this discussion is the appendix of the updated manuscript.
>
> Generally, the following equation shares the same probability path as both Eq. (7) and (9), for any $\alpha \in [0, 1]$:
> $$ \mathrm{d}x\_t = 2 \sigma'(t) \sigma(t) \nabla \log \left[p\_{\mathrm{data}} \star k\_{\mathrm{RBF}}^{\sigma(t)}\right] (x\_t) \mathrm{d}t - \alpha \sigma'(t) \sigma(t) \nabla \log \rho\_t(x\_t) \mathrm{d}t + \sqrt{2 \alpha \sigma'(t) \sigma(t)} \mathrm{d}W\_t. $$
> This corresponds to an interpolation between Eq. (7) and (9) that trades Brownian noise with its deterministic equivalent.
>
> This stochastic component can then be integrated in the formulation of interacting particle models, via Eq. (13). The latter equation takes the directions followed by the particles in the particle model, and transforms them via the operator $\mathcal{A}\_{\theta\_t}(z)$. Since this operator is linear, it is possible to integrate a stochastic component in the equation (Klebaner, 2012), allowing us to take into account stochastic particle models:
> $$ \mathrm{d}g\_{\theta\_t}(z) = [\mathcal{A}\_{\theta\_t}(z)]\left(\nabla h\_{\rho\_t} \mathrm{d}t + \gamma(t) \mathrm{d}W\_t\right), $$
> where $\gamma(t)$ is a scalar function of time.
>
> This makes it possible to integrate the stochastic component of diffusion models in Score GANs by interpolating between Gaussian noise and the score of the generated distribution in step 5 of Algorithm 1, similarly to the previous equation using $\alpha$. Nonetheless, this comes with no guarantee on the experimental performance, as we found in preliminary experiments that adding such a stochastic component is often detrimental to the resulting FID. Indeed, to succeed, the chosen gradient vector field to follow with the generator must be compatible with the generator architecture (i.e., compatible with the generator preconditioning $\mathcal{A}\_{\theta\_t}(z)$), which may not be the case with white noise of high variance.
>
> F. C. Klebaner. Introduction to stochastic calculus with applications. Imperial College Press. 2012.

---

> > ### Author Response · Authors · 2023-08-10
> > **Further Elements of Discussion for Reviewer 6P1y**
> >
> > The small character limit prevented us from completely answering the reviewer's many and detailed questions. We would like to offer additional elements of discussion in this message to start the discussion phase.
> >
> > ## Training of Score GANs
> >
> > The reviewer's interpretation is correct. This score network depends on the chosen noise level $\sigma$, like in EDM, but that we sample at each training step $t$ instead of scheduling it, unlike EDM. Indeed, $g_{\theta}(z)$ in Algorithm 1 corresponds to $g_{\theta_t}(z)$. The entirety of Algorithm 1 describes a single training iteration $t$.
> >
> > There are K steps of score updates per generator update, similarly to discriminators in GANs. Accordingly, $K$ is an important parameter in Score GANs. First of all, like in GANs, the tuning of $K$ heavily depends on the ratio $r = \frac{\lambda}{\eta}$ between the learning rates of the score network and the generator (Jelassi et al., 2022). A higher ratio may allow us to decrease the necessary number of steps $K$.
> >
> > In our experiments on image data, we use $r \geq 2$ (Appendix, Table 7). However, to gain more intuition empirically, we performed a set of experiments on MNIST with $\lambda$ s.t. $r = 1$ by making the number of steps $K$ vary from $1$ to $10$. We obtained similar results across this range of values for $K$, close to the values reported in Table 3. This indicates that even low values of $K$ can provide a sufficient approximation of the score of the generated distribution.
> >
> > To observe this qualitatively, we plotted in the document attached to our global response, Figure 3, the evolution of the score training loss in-between generator updates for $K=100$. At the beginning of training, $s_{\phi}^{\rho}$ needs around 20 updates to converge. However, after a low number of generator updates, it is already close to the optimum before its first update.
> >
> > This confirms that the continuous update of the score of the generated distribution $s_{\phi}^{\rho}$, like a discriminator, coupled with a learning rate ratio $r > 1$, makes a low number of score updates $K$ between generator updates sufficient.
> >
> > S. Jelassi et al. Dissecting adaptive methods in GANs. arXiv, 2022.
> >
> > ## Further Information on Discriminator Flows
> >
> > For time embeddings, we followed the implementation of EDM (Karras et al., 2023), which embeds time / noise levels using either Fourier random features (Vaswani et al., 2017) or positional encodings (Tancik et al., 2020).
> >
> > It is possible to train and infer the model using a second-order solver to solve Eq. (20) (step 2 of Algorithm 2). We experimented with this during the rebuttal period but did not observe a significant enough improvement in terms of NFE. Furthermore, we draw the reviewer's attention to our global response, where we explain that we must retract our claim that Discriminator Flows are more efficient than first-order EDM.
> >
> > A. Vaswani et al. Attention is all you need. NIPS 2017.\
> > M. Tancik et al. Fourier features let networks learn high frequency functions in low dimensional domains. NeurIPS 2020
> >
> > ## Discretization
> >
> > Studying how discretizing the considered continuous phenomena could affect our formulations is an interesting perspective for future work. We initiate a discussion on this topic below.
> >
> > Choosing the best discretization method for diffusion models is challenging (Karras et al., 2023): it depends on the chosen $\sigma(t)$, its efficiency is assessed w.r.t. NFE instead of the number of discretization steps like in numerical methods, and the final purpose of discretization (generating realistic data) differs from its initial one (approximating a solution to an SDE). Hence, standard approaches like EDM rely on empirical discretization grids and custom solvers, tailored to the generation task. Our framework, by identifying the true probability flow PDE of Eq. (9), may help diffusion models cope with discretization errors through the score of the generated distribution (as explained in line 109).
> >
> > The previous discussion, however, only holds for score-based diffusion models, for which the probability path is known in advance. For other particle models, this is not possible, and studying the convergence properties of their discretizations, like Arbel et al. (2019) do for MMD, is non-trivial.
> >
> > Adding a generator, as suggested, is an alternative way to solve the underlying particle model PDE. By generalizing the parallel between Wasserstein and Stein gradient flows in Section 3.3, generators can be seen in our framework as a preconditioning over the particle model PDE via the linear operator $\mathcal{A}_{\theta_t}(z)$ in Eq. (13). A well-chosen architecture, adapted to the particle flow $\nabla h$, may speedup and simplify the dynamics towards the data distribution.
> >
> >
> > ## Typos
> >
> > We will correct typos in the next version. The first part of Eq. (14) should indeed read as: $[\mathcal{A}\_{\theta\_t}(z)] (V) \triangleq \mathbb{E}\_{z' \sim p\_z} \left[k\_{g\_{\theta\_t}}(z, z') V(g\_{\theta\_t}(z'))\right]$.

---

### Official Review · Reviewer_iGU3 · 2023-07-04

**Soundness:** 3 good
**Presentation:** 3 good
**Contribution:** 3 good
**Rating:** 6
**Confidence:** 3

**Summary:**

The paper points out commonalities between diffusion and GAN models by examining the time evolution of the sample particles. For diffusion, this evolution happens at inference time by following the theory-given flow that transports the initial noise distribution to match the data distribution (in practice this flow is memorized by a neural network in the training stage). For GANs, it happens during training: at any given training step, the current generator induces positions for the particles, and they “slide down” the discriminator surface (which likewise jointly evolves to respond to the new positions). The paper formulates these modalities into a common framework of (interacting) particle models.

Based on this insight, the paper presents two practical algorithms, showing that either the discriminator or the generator can be removed and replaced with other diffusion-inspired mechanisms, while maintaining some GAN-like properties. The Score-GAN aims to follow the dynamics of a particle model in its training iterations by replacing the discriminator with an on-the-fly learned score that is combined with a pre-trained score of the noised data (as such, the model is not really a fully independent generative method). The Discriminator Flow drops the generator, and trains a time-varying discriminator, such that sliding initially white noise particles along its gradient brings them closer to the data.

**Strengths:**

I am not extensively familiar with the literature in this line of work, so I don’t have strong opinions about the magnitude of the novel insights in this paper. Personally I found it quite interesting and thought provoking, and as far as I can tell, theoretically sound.

The authors have put a good amount of effort into validating the theory and ensuing algorithms in practice. While the results are not high-quality by modern standards, the two models are somewhat intriguing, and could inspire further research. The combination of scores and reintroduced freedom for the network to decide on the paths, shape of the latent-to-image mapping, etc., might open up some interesting avenues, such as the speculated distillation use case.

The paper is polished and pleasant enough to read. However, there might be some miscellaneous errors in how the formulas are written (see Questions below), so going through to double check the rest of them once more could be helpful.

**Weaknesses:**

As acknowledged by the authors as well, the models are not really contenders for the state of the art at the moment. It’s possible (but by no means clear) that they could be significantly improved by a deep dive into engineering the design choices, but at the moment they mostly serve as a reasonable basic validation of the theory.

The Score-GAN approach seems to be limited in that it requires a pre-trained score function for the data. This seems to rule out the most common usage scenario of straight-out generation. Of course, one could argue that training that model is the first stage of the method, but at that point one could use it in standard diffusion. The actual algorithm seems to then essentially be distillation into a single-step method.

**Questions:**

How do latent interpolations (in z-space) with the proposed algorithms behave? Do you find a similar smooth morphing behavior as with standard GANs?

Algorithm 1 is, ultimately, a somewhat loose implementation of the theory: the estimated score is not trained to completion, expectation is evaluated stochastically, etc. I assume you also use something like Adam for training the generator with the update on Line 5 (?) Do you have any insight on whether the theory can accommodate some of these practicalities, or does it come down to intuition and empirics in the end? Are things like mode collapse a consequence of inexact adherence to theory?

Are these typos or such?
  - Eq. 14, should it be z’ in a couple of places instead of z?
  - Algorithm 1, you have x ~ p_data but then “overwrite” it with x = g(z), in two places – if I understood correctly, you shouldn’t be sampling from data distribution at all here?
  - Algorithm 1, line 5, again confusing wrt. line 4: x or x^sigma isn’t even used, alternatively it seems like you should instead have a noise addition on line 5 (like in appendix)
  - Algorithm 2 and Eq. 20, are there some gradient symbols missing around f? Isn’t it the gradient of c(f()) that you follow?

**Limitations:**

The paper acknowledges reasonably well various shortcomings of the methods, and the fact that the results are far from state of the art. There is a discussion of the broader impacts in the appendix.

---

> ### Author Rebuttal · Authors · 2023-08-09
>
> We would like to thank the reviewer for their feedback.
>
> We concur with the reviewer regarding the weaknesses raised, which are acknowledged in the paper. Notably, we mention that Score GAN can serve as a distillation method in the conclusion. We address the reviewer's questions below.
>
> ## Latent Interpolations
>
> We provide examples of interpolations on MNIST for all four tested models in the one-page PDF attached to our global response, Figure 2. We tested four interpolation methods on Gaussian priors considered by Leśniak et al. (2019): linear, spherical, Cauchy-linear, and spherical Cauchy-linear. We reached the same conclusion for each of these methods and thus show in the additional document the result of the most visually appealing one, spherical Cauchy-linear.
>
> We notice that the generator-based models, Score GAN and GAN, show smoother transitions between generated images than the particle models EDM and Discriminator Flow, for which abrupt changes of digit identity and shape can be seen between consecutive interpolation steps. This confirms that interacting particle models (generator-based) can perform feature learning via their smaller latent space, allowing for smoother generation, while particle models operating in the data space are less prone to such phenomenon. In this regard, Score GAN and Discriminator Flow are no different than their parent method in the same model category.
>
> We will add this discussion in the appendix of the next version of our paper.
>
> Leśniak et al. Distribution-Interpolation Trade off in Generative Models. ICLR 2019.
>
> ## Theory vs Practice in Algorithm 1
>
> We address the points raised by the reviewer below. As a preamble, we would like to notice that these remarks also hold for GANs, regarding the alternating updates of the discriminator and the generator, the use of Adam, and the stochasticity of expectation estimation via mini-batch training. This is then no suprise that Score GANs and GANs share properties like mode collapse, as pointed out in lines 285-286.
>
> ### Estimation of the Score
>
> > the estimated score is not trained to completion
>
> We would like to highlight that this is only true at the very beginning of training of Score GANs. Indeed, the score function is continuously optimized as its weights are carried on after each generator update, like a discriminator in GANs.
>
> To confirm this, we added an analysis of the training loss of the score function in the one-page PDF attached to our global response, Figure 3. In this experiment, we perform $K = 100$ discriminator steps in-between generator updates. At the beginning of training, the score function needs a few updates to converge. However, after a low number of generator updates, it is already close to the optimum before its first update.
>
> ### On Adam
>
> >  I assume you also use something like Adam for training the generator with the update on Line 5 (?) Do you have any insight on whether the theory can accommodate some of these practicalities, or does it come down to intuition and empirics in the end?
>
> We thank the reviewer for the interesting question. In fact, we do use Adam in practice. Theoretically, it's entirely possible to formulate continuous-time equations for Adam. By fixing the values of $(\beta_1, \beta_2)$ and allowing the time step to approach zero, we can recover the continuous version of SignSGD. A relevant example of SignSGD's study within a non-convex context was presented by Bernstein et al. (2018). Furthermore, exploring the scenario where non-interaction is present (where $\mathcal{A}_{\theta_t}$ disappears from Eq. (13) and described in lines 156-158) reveals a particle gradient flow with a renormalized gradient. This yields intriguing connections with continuous acceleration, as demonstrated by Wibisono & Wilsan (2015). We consider this avenue to be a promising direction for future investigation.
>
> A. Wibisono & A. C. Wilson. On accelerated methods in optimization. arXiv, 2015.\
> J. Bernstein et al. signSGD: Compressed optimisation for non-convex problems. ICML 2018.
>
> ### Other Factors and Mode Collapse
>
> > Do you have any insight on whether the theory can accommodate some of these practicalities, or does it come down to intuition and empirics in the end? Are things like mode collapse a consequence of inexact adherence to theory?
>
> While various factors, such as the inherent stochasticity of non-full-batch training, could contribute to mode collapse, we suppose that the primary cause lies in the departure from strict theoretical adherence. This departure occurs due to discretization errors introduced during the translation of the gradient flow into a discrete setting. This discrepancy between theory and practice could also be an interesting area for further exploration.
>
>
> ## Typos
>
> We would like to thank the reviewer for highlighting these typos and apologize for the mistakes.
> - The first part of Eq. (14) should indeed read as: $[\mathcal{A}\_{\theta\_t}(z)] (V) \triangleq \mathbb{E}\_{z' \sim p\_z} \left[k\_{g\_{\theta\_t}}(z, z') V(g\_{\theta\_t}(z'))\right]$.
> - All mentions of $x \sim p_{\mathrm{data}}$ in Algorithm 1 should be deleted; we sample from the generated distribution only.
> - For the two last typos, the reviewer is correct and they were already corrected in the PDF file of the supplementary material (where the main paper is reproduced).

---

> > ### Comment · Reviewer_iGU3 · 2023-08-18
> > **Response to rebuttal**
> >
> > Thank you for the thoughtful responses. I would would be happy with accepting the paper.

---

### Official Review · Reviewer_XiV8 · 2023-07-04

**Soundness:** 3 good
**Presentation:** 3 good
**Contribution:** 2 fair
**Rating:** 6
**Confidence:** 4

**Summary:**

The authors interpret GANs through the lens of particle optimization and show the links to score-based diffusion models. Based on these theoretical results, they propose to either 1) do particle learning (without a generator) based on the flow of a  discriminator, or 2) learn a generator by minimizing the score of the generator from the real data score.

**Strengths:**

They show interesting links between score-matching and particle-based GANs. They test some interesting ideas. They try different solvers for the particle-based GANs.

**Weaknesses:**

Particle GANs are not very novel; people don't use them frequently because its hard to optimize in high-dimensional space; this is not really new, but as far as I know, its rarely done, so this could still be interesting if it was really investigated deeply which they do not do (e.g., they don't verify the effect of different initializations, trying different optimizers, etc.). I know many papers study post-training flow/Langevin methods (e.g., Your GAN is Secretly an Energy-based Model and You Should use Discriminator Driven Latent Sampling), but this is slightly different from particle optimization. The score-based GANs might be novel, but I am not 100% certain.

Results are much worse for the disc-flow method compared to regular GANs and Diffusion and also the best GAN-Hybrid methods (e.g., Tackling the Generative Learning Trilemma with Denoising Diffusion GANs). It is well known that optimizing directly in the particle-space is difficult in high dimensions (I sadly do not remember those prior works, but I remember reading this), so this is not surprising.

Results are much worse for the ScoreGAN method compared to regular GANs and Diffusion.

Results are very limited, with only MNIST and CelebA and 2 baseline methods. Maybe the proposed approaches are worse because the architectures are not that good, or maybe minimizing the score-function loss is not enough; more in-depth analyses using different optimizers, initializations for the particles, and architectures on slightly bigger datasets would add a lot.

In the related work, I would consider adding papers that combine GANs with diffusion or score-based models. (These are some I could come up with, but a few more exist)
Diffusion non-score-based:
- Diffusion-GAN: Training GANs with Diffusion (2022)
- Tackling the Generative Learning Trilemma with Denoising Diffusion GANs (2022)
Score-based:
- Adversarial score matching and improved sampling for image generation (2020)

Overall: Weak results, limited novelty, limited experimentation to try to make these approach perform better than their current performance (worse than every baseline methods).


**Questions:**

line 21: 'challenge the conventional view that particle and adversarial generative models are opposed to each other': the Discriminator can be used to learn particles (learn x_fake directly); this is well known as far as I know.

'Score GANs': I would encourage trying to find another name that is a bit more explicit about your method; it's a bit too generic, and other papers have used the same name.

I mentioned some things like extended analyses in the 'Weakness' section.

**Limitations:**

The only limitation mentioned is that the generator-less method is slower. They don't mention many limitations. They say the results may be poor because they are too novel and have yet to be fine-tuned. I don't consider those enough.

---

> ### Author Rebuttal · Authors · 2023-08-09
>
> We would like to thank the reviewer for their feedback. We address the weaknesses they raised below.
>
> ## Novelty
>
> > Particle GANs are not very novel; [...] this is not really new, but as far as I know, its rarely done
>
> > line 21: 'challenge the conventional view that particle and adversarial generative models are opposed to each other': the Discriminator can be used to learn particles (learn x_fake directly); this is well known as far as I know.
>
> Our understanding is that the reviewer questions the novelty of Section 3.2 and of the introduced Discriminator Flow model. However, they did not provide any reference to support their assertion, so we have no choice but to respectfully maintain our claims. We welcome any suggestion to improve our positioning w.r.t. related work that we may have missed.
>
> Moreover, we would like to point out that we did appropriately cite and build upon prior works studying the particle model aspect of GANs, albeit not as general as our approach: e.g., Franceschi et al. (2022) on line 162, Yi et al. (2023) on line 183, as well as Chu et al. (2020) and Durr et al. (2022) on line 196.
>
> In particular regarding the novelty of Discriminator Flows, the closest model we are aware of is the one of Heng et al. (2023). We notice in lines 305-306 that their model is a particular case of Discriminator Flows, although the original article did not make the link with discriminators and GANs.
>
> ## Experiments and Performance
>
> > Results are much worse for the disc-flow [and the ScoreGAN] method compared to regular GANs and Diffusion and also the best GAN-Hybrid methods [...].
>
> > Results are very limited, with only MNIST and CelebA and 2 baseline methods. [...] more in-depth analyses using different optimizers, initializations for the particles, and architectures on slightly bigger datasets would add a lot.
>
> We understand that better performance and more thorough experiments could only improve the paper and support our framework. However, the reviewer's focus on state-of-the-art experiments differs from the paper's scope which is mostly theoretical. In our support, we recall the following statement from Reviewer iGU3:
> > **Reviewer iGU3:** *As acknowledged by the authors as well, the models are not really contenders for the state of the art at the moment. It’s possible (but by no means clear) that they could be significantly improved by a deep dive into engineering the design choices, but at the moment they mostly serve as a reasonable basic validation of the theory.*
>
> ## Optimization in the Particle Space
>
> > people don't use [particle GANs] frequently because its hard to optimize in high-dimensional space
>
> > It is well known that optimizing directly in the particle-space is difficult in high dimensions (I sadly do not remember those prior works, but I remember reading this)
>
> We respectfully disagree with the reviewer. We know that gradient descent in particle space is not inherently flawed: after all, diffusion models are also particle models and they are state-of-the-art in image generation (Dhariwal & Nichol, 2021). Furthermore, Heng et al. (2023) (with their model which we show is a special case of Discriminator Flows) and Fan et al. (2022) also present particle models with competitive performance.
>
> It is indeed known that *some* particle models suffer from the the curse of dimensionality, but not all. How the curse of dimensionality affects gradient descent of a functional in the particle space is closely related to how the curse of dimensionality affects the functional itself. Indeed, if the sample complexity of the functional scales badly in the dimension, it is expected that the resulting gradient flow also inherits this curse. A prototypical example of this is given by the 1-Wasserstein distance (Dudley, 1969), which the reviewer may have thought about. However, in our case, the functional is (implicity) implemented by a neural network, known to be particularly effective in high dimensions. This explains why particle models such as ours can be effective even in high dimensions.
>
> R. M. Dudley. The speed of mean Glivenko-Cantelli convergence. Ann. Math. Statist. 1969.
>
> ## Related Work
>
> The reviewer suggests that we discuss the following papers:
> 1. Wang et al. Diffusion-GAN: Training GANs with diffusion. arXiv, 2022.
> 2. Xiao et al. Tackling the generative learning trilemma with denoising diffusion GANs. ICLR 2022.
> 3. Jolicoeur-Martineau et al. Adversarial score matching and improved sampling for image generation. ICLR 2021.
>
> We thank the reviewer for the suggestion. We would like to point out that **we already cite 1. in Section 4.4, line 299** as strongly linked to Score GAN. **We also cite 2. in our introduction**. We will further discuss 2. and 3. in the next version of the manuscript.
>
> More specifically, 2. and 3. are not directly related to our unified view of score-based diffusion and GANs as particle models. 2. trains several GANs to successively denoise an image, similarly to diffusion. 3. augments the denoising objective of score-based diffusion models with an adversarial objective to improve the denoising image quality. None of these references leverage the link we make between GANs and particle models.
>
> ## Limitations
>
> > The only limitation mentioned is that the generator-less method is slower. They don't mention many limitations. They say the results may be poor because they are too novel and have yet to be fine-tuned. I don't consider those enough.
>
> As most of the reviewers noticed (AQyb, HYQ5, iGU3 and 6P1y), we addressed the limitations of our approach in the paper, including but not limited to the ones the reviewer stated in this quote. We would welcome any suggestion of additional limitations to discuss from the reviewer.
>
> Beyond this, we would also like to point out that we addressed potential negative impacts of our work in the supplementary material, Appendix Section D.

---

> > ### Comment · Reviewer_XiV8 · 2023-08-10
> >
> > The authors rebuttal is very convincing. I agree that there is very little other papers discussing particle GANs. This is mostly a theory paper, not an empirical one. For Neurips, I would still except stronger experiments, but nevertheless it deserve reach for its interesting theory and future potential. I appreciate that they addressed all my questions.
> >
> > I don't think that I can edit the score directly, it doesn't allow me. But I would now rescore it as a 6/10.

---

### Official Review · Reviewer_kQU7 · 2023-07-08

**Soundness:** 3 good
**Presentation:** 2 fair
**Contribution:** 3 good
**Rating:** 5
**Confidence:** 2

**Summary:**

The paper proposes a framework that unifies particle-based deep generative models, such as gradient flows and score-based diffusion models, and adversarial generative models, such as GANs, by framing generator training as a generalization of particle models. The authors suggest that a generator is an optional addition to any such generative model and show how it is possible to train a generator with score-based gradients which replace adversarial training and to train a GAN without a generator, using only the discriminator to synthesize samples. The paper presents two new models, a Score GAN and a Discriminator Flow, as proofs of concept to illustrate the viability of these ideas.

**Strengths:**

- Overall, the paper is well-organized and well-written.
- This paper investigates the two most prominent classes of generative models in the field, i.e., GANs and diffusion models, and provides a unified framework that explains their similarities. This work is valuable as it has the potential to inspire future research efforts to unify these two classes of generative models in addressing their respective problems.
- This paper has a relatively strong theoretical foundation and meets the standards of the conference.

**Weaknesses:**

- Some mathematical symbols (e.g., $\nabla_W$) require further explanation to enhance the readability of this paper for a wider range of readers.
- The experiments in this paper are relatively weak, with a limited number of datasets and tests conducted. Furthermore, according to Table 3, the proposed methods (Discriminator Flow and Score GAN) fail to surpass baseline methods (EDM and GAN).

**Questions:**

- In line 29, what does it mean that the generator has the role of a smoothing operator?
- What is the time cost of training Score GAN and Discriminator Flows? Do they have a greater training overhead than Score-based Model and GAN?

**Limitations:**

No, the authors have not adequately addressed the limitations and potential negative societal impact of their work.

---

> ### Author Rebuttal · Authors · 2023-08-09
>
> We thank the reviewer for their feedback. We address the weaknesses and questions they raised below.
>
> ## Wasserstein Notation
>
> > Some mathematical symbols (e.g., $\nabla_W$) require further explanation to enhance the readability of this paper for a wider range of readers.
>
> The notation $\nabla_W$ is a standard notation for readers with expertise in Wasserstein gradient flows, but it may indeed be confusing for a wider range of readers. For a functional $\mathcal{F}$ defined over probability measures, $\nabla_W \mathcal{F}(q)$ is the so-called Wasserstein gradient of the functional -- similar to the gradient of a functional defined over a Euclidean space. It is defined as $\nabla_x \frac{\delta \mathcal{F}}{\delta q}$ (Eq. (3)), i.e. the Euclidean gradient of the functional’s first variation $\frac{\delta \mathcal{F}}{\delta q}$. We refer to Santambrogio (2017), cited in the paper, for more information.
>
> ## Experimental Results
>
> > The experiments in this paper are relatively weak, with a limited number of datasets and tests conducted. Furthermore, according to Table 3, the proposed methods (Discriminator Flow and Score GAN) fail to surpass baseline methods (EDM and GAN).
>
> Providing state-of-the-art performance in our experimental results is beyond the scope of our paper. We include the experimental results in our paper primarily for proof-of-concept purposes, to validate that our proposed Discriminator Flow and Score GAN models are practically feasible. Please also refer to our global answer on this topic.
>
> ## Generator as a Smoothing Operator
>
> > In line 29, what does it mean that the generator has the role of a smoothing operator?
>
> This role of the generator is formally defined and explained in Definition 2 (lines 148-151). We note that compared to the particle model (PM) given in Definition 1 and Equation 1, the use of a generator results in applying a linear operator on top of the vector field $\nabla h_{\rho_t}$. It is this linear operator role that provides the intuition that the generator acts as a smoothing operator (analogous to a smoothing filter). Indeed, we notice in line 151 that this linear operator is a kernel integral operator, which by design smoothes the input signal $V$ with a kernel convolution in Eq. (14).
>
> ## Training Costs
>
> > What is the time cost of training Score GAN and Discriminator Flows? Do they have a greater training overhead than Score-based Model and GAN?
>
> Indeed, the design of Score GAN and Discriminator Flow induces computational constraints that make each of their training iterations slower than those of baseline models.
>
> Score GAN requires pretraining a score network as specified in the paper, and its score-based update remains computationally more demanding than a discriminator-based update like in GANs (since the score function takes values in the data space, while the discriminator output is scalar).
>
> As explained in lines 270 and 287, training Discriminator Flow requires sampling at every step from the generating ODE of Eq. (20). This makes its training iterations slower than both diffusion models (which do not require resampling through the ODE/SDE) and GANs (which have fast sampling).
>
> Besides the cost of individual training iterations, the total temporal cost of training also depends on the number of iterations which we specify in Appendix C.2. Overall, the approximate amount of time we used to train the tested models over MNIST on one NVIDIA V100 GPU are:
> - for Discriminator Flow, 24 hours;
> - for Score GAN, 10 hours (excluding the pretrained diffusion model);
> - for EDM, 6 hours;
> - for GAN, 1 hours.
>
> We will highlight the elements of this discussion in the updated version of the manuscript.
>
> ## Limitations and Societal Impact
>
> > No, the authors have not adequately addressed the limitations and potential negative societal impact of their work.
>
> We would like to point out that **we did address potential negative impacts** of our work in the supplementary material, Appendix Section D.
>
> Furthermore, **we did address limitations** of our approach throughout the paper, including:
> - the generative performance of the introduced models, lines 279 to 282;
> - the training cost efficiency of the introduced models, lines 241, 270 and 287;
> - the omission of continuous training of the GAN discriminator, lines 169 and 342-343.
>
> Nonetheless, we welcome specific suggestions from the reviewer to improve this part of the paper.

---

### Official Review · Reviewer_HYQ5 · 2023-07-26

**Soundness:** 3 good
**Presentation:** 3 good
**Contribution:** 3 good
**Rating:** 7
**Confidence:** 2

**Summary:**

This paper proposes a framework, i.e., a generalization of particle models, that unifies particle-based generative models like score-based diffusion and gradient flows with adversarial generative models like GANs, where generator training is framed as interacting Particle Models. Based on this perspective, the paper introduces two new hybrid models Score GANs (using score model to train generator) and Discriminator Flows (as prooof of concept. Experiments on MNIST and CelebA confirm the viability of the new framework.

**Strengths:**

+ The findings from mathematical derivation seem to be interesting.
+ The writings are mostly clear and sound, well-supported by mathematical derivations and experiments.

**Weaknesses:**

+ The claim that "Discriminator Flows have the expected advantage of converging faster to the target distribution as compared to the
state-of-the-art diffusion model EDM, and thus have higher efficiency during inference." may be overstated, as s Figure 5 in the Appendix demonstrates that EDM (Heun solver) exhibits comparable efficiency. Which solver is employed for EDM in Figure 1?

**Questions:**

+ Why both $x \sim p_{data}$ and $x = g_θ(z)$ exists in Algorithm 1?

**Limitations:**

Yes, the authors talked about the limitations of models proposed, and I believe these models are more of proof of concept. The authors also discussed social impact in Appendix.

---

> ### Author Rebuttal · Authors · 2023-08-09
>
> We thank the reviewer for their feedback. We address the questions they raised below.
>
> > The claim that "Discriminator Flows have the expected advantage of converging faster to the target distribution as compared to the state-of-the-art diffusion model EDM, and thus have higher efficiency during inference" may be overstated, as Figure 5 in the Appendix demonstrates that EDM (Heun solver) exhibits comparable efficiency. Which solver is employed for EDM in Figure 1?
>
> In Figure 1, we used the Heun solver for EDM. However, Figure 1 does not correspond to the setting of Figure 5. In Figure 5, we studied two different ways to reduce the number of NFE with Discriminator Flows: by discretizing the generating differential equation of Eq. (20) -- Figure 5(a) -- and by stopping the generation process early, like in Figure 1 -- Figure 5(b). We compared these two versions to the canonical way of reducing NFE in a score-based diffusion model, i.e. by discretizing the generating differential equation of Eq. (8) like Karras et al. (2022), either with Euler or Heun solver. Those are the two baselines identically reproduced in both Figures 5(a) and 5(b).
>
> We included in the one-page PDF attached to our global response, in Figure 1(a), the merged plots of Figures 5(a) and 5(b). To compare directly performance vs NFE in the same setting as in Figure 1 in the paper, we evaluated EDM in early stopping similarly to Discriminator Flows in the one-page PDF attached to our global response, Figure 1(b). As expected, diffusion models need to completely denoise the image to achieve reasonable performance.
>
> Nevertheless, after noticing an unrelated mistake in our experimental setting, we unfortunately have to retract the claim questioned by the reviewer. Please refer to the global response for more details.
>
> > Why both $x \sim p_{\mathrm{data}}$ and $x = g_{\theta}(z)$ exists in Algorithm 1?
>
> We apologize for the mistake. $x \sim p_{\mathrm{data}}$ should not appear in Algorithm 1. We will correct this in the updated version of the manuscript.

---

> > ### Comment · Reviewer_HYQ5 · 2023-08-18
> >
> > Thanks for the reply. I will keep my score.

---

### Official Review · Reviewer_AQyb · 2023-07-28

**Soundness:** 3 good
**Presentation:** 3 good
**Contribution:** 3 good
**Rating:** 7
**Confidence:** 3

**Summary:**

This work presents a unified generative particle model framework that explains generative adversarial networks (GANs) and score-based diffusion models.
This model is characterized by an evolution equation with a functional  ${h_{\rho}}_t$ that captures the transitioning over the time index $t$.
This work explains that the generation/inference of score-based diffusion models and the training of generators for GANs follow this model.
These particle models allow the decoupling of gradient flow in the score-based models and the generator in the GANs.
This led to the two new models:
(1). Scored-based GANs with generators that trained with score-based gradients instead of using adversarial learning.
(2). GANs that train without a generator. Only the discriminator synthesizes samples.


**Strengths:**

The paper is well-organized.
It also cited relevant works in a way that makes it easy to understand the main point.
The work is also important because it offers a new perspective into the workings of two main classes of generative AI models.
This opens the door to understanding and ultimately making beneficial adjustments to these models.

**Weaknesses:**

Some of the underlying assumptions and claims are not clear.
The particle model for score-based models represents the generation/inference of the models.
The interacting particle model for generator-based models is the training of the generator.
It is not clear if the functional $h_{\rho}$ in both cases are interchangeable.

Also, it is not clear how you arrived at Finding 6.
Please provide more details on the equivalence of the interacting particle model of GANs and the diffusion models.




**Questions:**

(1). The following phrase is not clear:  Line 51 on Page 2:  "In prior work, ...". It is not clear if this is referring to a specific work or prior works.

(2). It is not obvious that the modeling of the generation/inference with score-based models is interchangeable with training generators. Please can you provide more information on this?

(3). Please provide more details on the equivalence of the interacting particle model of GANs and the diffusion models as stated in Finding 6.

(4). Is it possible to come up with a hybrid structure that combines the two main techniques under the unified PM model?  If yes, do you think this could improve the performance?

**Limitations:**

The author(s) addressed the limitation of their work.

---

> ### Author Rebuttal · Authors · 2023-08-09
>
> We thank the reviewer for their feedback. We address the weaknesses they raised by answering their questions below.
>
> ## (1). Wording Clarification
>
> > (1). The following phrase is not clear: Line 51 on Page 2: "In prior work, ...". It is not clear if this is referring to a specific work or prior works.
>
> This is referring to prior works, plural. We meant that the choice of a prior that is easy to sample from is a widely adopted choice in the literature.
>
>
> ## (2). and (3). Equivalence Between Particle Models and GANs
>
> > (2). It is not obvious that the modeling of the generation/inference with score-based models is interchangeable with training generators. Please can you provide more information on this?
> > (3). Please provide more details on the equivalence of the interacting particle model of GANs and the diffusion models as stated in Finding 6.
>
> We do not claim that particle model (PM) inference and generator training are equivalent or interchangeable. Instead, we show that interacting particle models (Int-PMs), and in particular GANs, ***generalize*** PM inference during generator training, as worded in Findings 4 and 6. Indeed, the particle evolution of an Int-PM (Eq. (13)) reduces to the particle evolution of a PM (Eq. (1)) with the same functional $h_{\rho}$ when $k_{g_{\theta_t}}(z, z') = \delta_{z - z'}$ (lines 152 - 158), in which case $\mathcal{A}_{\theta_t}(z)$ disappears from Eq. (13) to obtain Eq. (1).
>
> This indicates that a functional $h_{\rho}$ used in an Int-PM may be used in a PM and vice versa. However, note that this does not guarantee successful performance, as the compatibility of the generator with $h_{\rho}$ (through $\mathcal{A}_{\theta_t}(z)$) should also be taken into account.
>
> We confirm the possibility of transferring $h_{\rho}$ with the models we introduce: the PM Discriminator Flow is based on an Int-PM GAN functional, and the Int-PM Score GAN is based on an PM score functional.
>
> Furthermore, this is supported by Finding 6 where we notice that functionals $h_{\rho}$ from typical PMs have been shown to be encompassed in the Int-PM GAN framework. Under some hypotheses on the discriminator, Yi et al. (2023) show that the $h_{\rho}$ of $f$-divergence GANs is actually the functional of the forward KL divergence gradient flow (like in the NCSN diffusion model of Song et al. (2019)), and Franceschi et al. (2022) show that it is the one of the squared MMD gradient flow. Considering Finding 4, this therefore means that GANs generalize these PMs.
>
> Nonetheless, we agree that the formulation of Finding 6 should be more specific. We will clarify our explanations in Section 3 and specify exactly which score models and Wasserstein gradient flows are generalized by GANs based on the previous paragraph.
>
> ## (4). Hybrid Models
>
> > (4). Is it possible to come up with a hybrid structure that combines the two main techniques under the unified PM model? If yes, do you think this could improve the performance?
>
> The introduced Score GAN and Discriminator Flow in Section 4 are examples of combinations of such techniques, as explained above: the PM Discriminator Flow is based on an Int-PM GAN functional, and the Int-PM Score GAN is based on an PM score functional. However, they are only proof-of-concept examples. As highlighted in the conclusion, we believe that finding a hybrid model that outperforms both diffusion and GANs is an interesting direction for future work regarding our framework.

---

> > ### Comment · Reviewer_AQyb · 2023-08-17
> > **Response to Rebuttal.**
> >
> > Thank you for responding to my questions.  Your response clarifies most of the issues I raised. Consequently, I will be upgrading my rating.

---

### Author Rebuttal · Authors · 2023-08-09

## Response to Reviews

We would like to thank the reviewers for their insightful, constructive, and timely reviews. We are glad that most reviewers appreciated the paper and the perspective we propose, generally agreeing on its relevance and potential for the community.

Below, we respond to each reviewer individually. The readers will find attached to this global response a one-page PDF of figures illustrating some of our discussions, including interpolations in the latent space and an analysis of the training dynamics of Score GANs.

A recurring concern raised by the reviewers deals with the experimental results. We understand that better performance and more thorough experiments could improve the paper and provide further support to our framework. Nonetheless, we would like to remind the reviewers that the proposed models and experiments serve as a "reasonable basic validation of the theory" (Reviewer iGU3), given the more theoretical focus of the paper, as also noted by Reviewers HYQ5 and 6P1y.

Some reviewers spotted typos throughout the paper. We would like to point out that we had already corrected the most important ones in the appendix provided in the supplementary material. We refer the reviewers to this document for a more polished version of the paper.

We look forward to further discussion with the reviewers during the discussion period.

## Correction: Appendix B.3

During our work on this rebuttal, we noticed a mistake in the experiments in Appendix B.3, which changes the conclusion that Discriminator Flows are more efficient in terms of NFE than the diffusion model EDM with a first-order solver. Thus, we report this correction to the reviewers.

In the original submission, the results of the "EDM (Euler solver)" baseline in Figure 5 were computed for the stochastic version of EDM instead of the deterministic one like in the rest of the paper. The corrected figure with the deterministic baseline can be found in the attached one-page PDF, Figure 1(a).

Surprisingly, the first-order version of EDM is actually more efficient in terms of NFE than the second-order version for low NFE values. This baseline was not considered in the EDM paper (Karras et al., 2022), so this is a new observation. We confirmed this phenomenon using the official implementation of EDM on CIFAR10.

Unfortunately, this invalidates our initial claim that Discriminator Flows are more efficient that the first-order EDM with the chosen discretizations. We still expect that, with proper tuning, Discriminator Flows can be more efficient than diffusion models, based on Figure 1 of our paper -- which remains valid. The final image appears quickly after a few steps, but some residual noise still needs to be eliminated in the remaining steps.

Nevertheless, this result was secondary and all other results of the paper remain valid, as this mistake only impacted this experiment. We apologize for the confusion and thank the reviewers in advance when taking these new elements into account.

---

### Decision · Program_Chairs · 2023-09-21

**Decision:**

Accept (poster)

**Comment:**

2x A, 2x WA, and 2x BA. This paper proposes to unify GANs and score-based diffusion models through particle optimization. The reviewers agree on accepting the paper due to its (1) clear presentation, (2) important topic, and (3) sound theoretical derivation. The rebuttal has addressed their concerns.